# DiasR: Dual-Modal Identity-Anchored Sparse Routing for Efficient Multi-Subject Video Generation

**Yangyang Li**[1 2]  **Wu Liu**[1]  **Jie Li**[3]  **Xinchen Liu**[4]  **Yongdong Zhang**[1]  **Guoqing Jin**[2]

## Abstract

Personalized multi-subject video generation is a promising direction within the field of controllable video generation; however, existing methods face challenges in maintaining cross-frame identity consistency and incur high computational overhead. To address these issues, we propose DiasR, an efficient framework that integrates Dual-Modal Identity-Anchored Alignment and a novel Sparse Routing Strategy. The Dual-Modal Identity-Anchored Alignment employs learnable identity queries to align visual and textual modalities with ground-truth subject masks, thereby mitigating cross-frame identity drift. The Sparse Routing Strategy dynamically routes video tokens to relevant subjects and groups them through bucket aggregation, reducing computational overhead and alleviating identity entanglement induced by redundant tokens. We also construct MuSA-2M, a large-scale dataset comprising 2 million annotated samples equipped with subject-level masks, which fills the gap in existing multi-subject video datasets. Experiments conducted on the OpenS2V-Eval benchmark demonstrate that our method achieves superior performance in identity consistency, text fidelity, and video naturalness. Notably, it maintains a nearly constant inference time as the number of reference subjects increases, outperforming existing baselines in both efficiency and generation quality for scenarios involving multi-subject interactions. Project Page: https://tale17.github.io/diasr/.

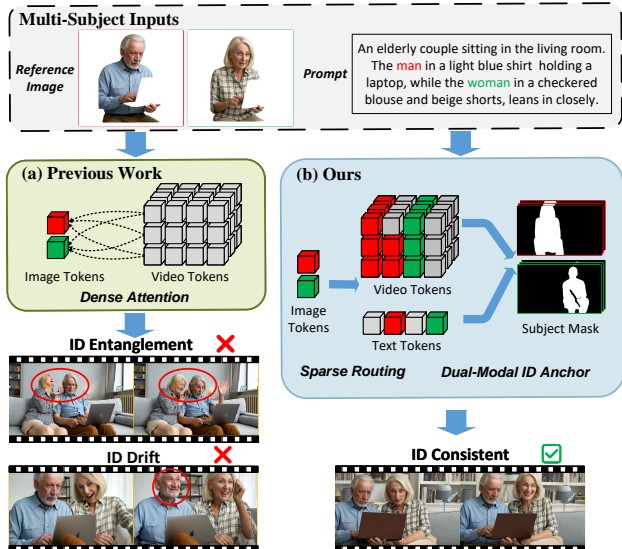

*Figure 1.* **(a)** Methods relying on dense attention suffer from spatial identity entanglement and temporal identity drift. **(b)** Our sparse routing strategy between subjects and video tokens enables the model to learn independent subject representations via dual-modal identity anchors, thereby achieving spatiotemporal consistency.

## 1. Introduction

With the recent advances in multimodal agent system (Liu et al., 2026) and diffusion transformers (DiTs) (Peebles & Xie, 2023) for video generation (Yang et al., 2025; Wan et al., 2025; Zheng et al., 2024; Lin et al., 2024; Kong et al., 2024), personalized multi-subject video generation has emerged as a promising direction in controllable video generation. Given a text prompt and a set of reference images, it aims to enable filmmakers, advertisers, and content creators to automatically produce personalized scenes featuring rich inter-subject interactions. Reliable spatio-temporal alignment between text prompts and visual subjects is the core prerequisite for such tasks (Liu et al., 2024). However, current methods seldom achieve both (i) cross-frame identity consistency for subjects and (ii) physically and semantically plausible inter-subject interactions, which substantially limits their applicability in real-world multi-subject scenarios.

In recent years, several studies (Liu et al., 2025b; Fei et al., 2025; Jiang et al., 2025; Deng et al., 2025; Wang et al.,

---

[1]School of Information Science and Technology, University of Science and Technology of China, Hefei, Anhui, China [2]The State Key Laboratory of Communication Content Cognition, People's Daily Online, Beijing, China [3]Zhejiang University, Hangzhou, Zhejiang, China [4]JD Explore Academy, JD.com Inc., Beijing, China. Correspondence to: Guoqing Jin <jinguoqing@people.cn>, Xinchen Liu <liuxinchen1@jd.com>.

*Proceedings of the 43rd International Conference on Machine Learning*, Seoul, South Korea. PMLR 306, 2026. Copyright 2026 by the author(s).

2025b;a; Liang et al., 2025; He et al., 2025; Hu et al., 2025; Chen et al., 2025) leverage diffusion transformers (DiTs) to unify visual representation and temporal modeling, integrating multiple reference images as appearance priors for a single video output. Concretely, these studies encode reference images using the base model's VAE to generate latent tokens, concatenate these tokens with the noisy video token sequence, and inject subject identities into the video through attention mechanism. By leveraging the high-dimensional spatiotemporal generalization capability of DiTs, these methods require minimal modifications to the base model and learn visual conditional injection via in-context self-attention, thus enabling multi-subject identity injection. However, such context-driven approaches only model the data distribution in an aggregated manner and lack subject-specific consistency control, rendering them prone to temporal identity drift and spatial identity entanglement ( Figure 1). Consequently, they rely heavily on the quality and diversity of datasets. Moreover, the additional sequence length elevates computational cost, as the attention overhead of DiT scales quadratically with it.

In this paper, we present DiasR, a framework for personalized multi-subject video generation which is based on **D**ual-Modal **I**dentity-**A**nchored alignment and a **S**parse **R**outing strategy. To enhance subject-level perception in both visual and textual modalities, we introduce a series of learnable identity queries, each anchoring a distinct subject. We project these queries onto the visual and textual spaces and fuse them with the corresponding subject-specific tokens. During the attention mechanism, we compute dual-modal attention perceptions between the noisy video latent features and the identity-anchored token streams and explicitly align these perceptions with groundtruth subject masks. This dual-modal perception alignment facilitates subject-specific identity learning, thereby mitigating identity drift, while simultaneously improving prompt-level controllability of each subject in multi-subject scenarios.

Furthermore, to mitigate the quadratic computational growth caused by additional tokens in multi-subject scenarios, we propose a sparse-routing strategy. Prior to attention computation, we dynamically route each video token to its most relevant subject and regroup them through bucket aggregation. For $N$ subjects, each with a sequence length of $lh \cdot lw$, this strategy eliminates $(N-1) \cdot lh \cdot lw$ redundant subject tokens relative to naive concatenation (amounting to approximately 10% of all tokens), thereby converting variable-length multi-subject token sequences into a constant-length representation and reducing memory usage and wall-clock time costs. In addition, by avoiding the injection of multiple subject features into the same spatial region, the strategy alleviates subject entanglement in the result.

Additionally, we develop a scalable data curation pipeline to collect multi-subject video data accompanied by reference images and corresponding subject-mask annotations, filling the gap in subject-level masks within existing video datasets. Leveraging this pipeline, we curate a Multi-Subject Annotated Video Dataset (MuSA-2M) comprising 2 million samples spanning three key interaction scenarios: human–human, human–object, and multi-object interactions, each annotated with detailed subject information. With these components, our framework delivers high-fidelity and computationally efficient multi-subject video generation.

We summarize our main contributions as follows:

- We construct **Multi-Subject Annotated Video Dataset** (MuSA-2M), a novel large-scale dataset with subject-level mask annotations, tailored to the task of personalized multi-subject video generation.

- We introduce **Dual-Modal Identity-Anchored Alignment**, a mechanism that aligns the dual-modal attention perceptions to each subject via ground-truth-based supervision, thereby enhancing identity consistency and text fidelity in multi-subject scenarios.

- We propose **Sparse Routing Strategy** that reduces computational overhead associated with additional subject tokens and mitigates identity blending by preventing redundant injections of multi-subject embeddings.

## 2. Related Work

### 2.1. Subject-driven Video Generation

Subject-driven Video Generation (S2V) aims to generate text-prompt-aligned videos while preserving subject identity. Prior approaches (He et al., 2024; Yuan et al., 2025) extract visual features via multimodal encoders (Radford et al., 2021) and inject them into Diffusion Transformers (DiT) through adapters or cross-attention, but still suffer from limited identity consistency. Recent works also explore identity-aware attention control and layout-guided generation. CustomVideo (Wang et al., 2026) constrains textual attention maps to associate identities with textual tokens, while ContextGen (Xu et al., 2026) introduces predefined attention regions for controllable multi-subject layout generation. However, these methods can not establish explicit dual-modal identity alignment thus less effective for preserving precise human identities. HumanNeRF-SE (Ma et al., 2024a) improves pose-driven animation with diverse pose control, but they focus on single-subject reconstruction and animation rather than multi-subject video generation.

Insights from Subject-driven Image Generation (S2I) methods (Tan et al., 2025; Xiao et al., 2025; Mao et al., 2025; Mou et al., 2025) show that VAE-extracted visual features (Kingma & Welling, 2014), which share the same

latent space as video latents, better preserve subject details and ensure identity consistency. These features are concatenated with noisy token sequences for in-context conditioning, enabling strong subject control without extensive base model modifications. Building on this, methods like FullDiT (Ju et al., 2025), SkyReels-A2 (Fei et al., 2025), and HuMo (Chen et al., 2025) integrate visual control via full attention; MAGREF (Deng et al., 2025) aggregates multiple subjects for full attention; HunyuanCustom (Hu et al., 2025) and PolyVivid (Hu et al., 2026) use LLMs for prompt embeddings; InteractHuman (Wang et al., 2025b) employs subject masks; and Kaleido (Zhang et al., 2025b) enhances consistency via reference rotation encoding. Saber (Zhou et al., 2025) focuses on scaling zero-shot S2V by removing the dependence on dedicated triplet training data, introducing mask-based dynamic references to simulate subject-driven training for general subject consistency. While achieving high-fidelity S2V, these methods often face identity drift or subject blending in multi-subject scenarios due to lack of explicit identity distinction. In contrast, our method introduces a dual-modal identity-anchored alignment mechanism to explicitly distinguish and align each subject, improving multi-subject identity consistency.

## 2.2. Sparse Computation in Diffusion Transformers

The computational cost of DiT scales quadratically with the token sequence length, rendering it inefficient for extremely long sequences. To alleviate the degradation in computational performance caused by long contexts, some previous works accelerate computation via quantization and pruning (Li et al., 2023; Castells et al., 2024), or by reducing the number of sampling steps (Ma et al., 2024b; Chen et al., 2024; Selvaraju et al., 2024). Other studies analyze the attention mechanism in DiT and observe that attention matrices are typically sparse with substantial redundancy. Building on this observation, SVG (Xi et al., 2025) exploits the inherent spatiotemporal sparsity of attention heads to identify dynamic sparse patterns online, thereby removing redundancy without the need for retraining. SVG-2 (Yang et al., 2026) further leverages semantic-aware permutation via k-means clustering and incorporates dynamic top-k critical-token selection. SparseAttention (Zhang et al., 2025a) predicts attention maps to skip selected matrix blocks and applies quantization to specific operators. HiStream (Qiu et al., 2025) achieves token-level sparsity by compressing spatiotemporal redundancy based on static priors and pruning weak attention correlations via a sliding window. OneStory (An et al., 2026) adopts semantics-guided frame-level sparsity for multi-shot video generation by selecting top-K semantically relevant historical frames and adaptively compressing the context. MoC (Cai et al., 2026) achieves multi-shot long-video generation by employing full attention within shots and sparse routing across shots. However, these methods pri-

marily target sparsity-driven acceleration for long-sequence video generation and lack routing safeguards between the video sequence and the reference sequence in S2V tasks, which can compromise identity consistency; moreover, excessively sparse video sequences can degrade generation quality. FullDiT-2 (He et al., 2025) improves efficiency by compressing token-level redundancy in the video context, using an MLP to estimate token importance from Value features. However, it overlooks semantic associations between video tokens and specific subjects, which may lead to multi-subject entanglement. In contrast, our approach introduces a dynamic subject-level sparse routing strategy whereby each video token is routed to its most relevant subject, enabling sparse acceleration while preserving identity consistency.

## 3. Method

To address identity drift, identity entanglement and excessive computational overhead in multi-subject video generation, we propose **DiasR**, a **D**ual-Modal **I**dentity-**A**nchored **S**parse **R**outing framework for efficient, identity-consistent multi-subject video generation. DiasR takes reference images $\{I_i\}_{i=1}^N$ and a text prompt as inputs and efficiently generates videos with persistent subject identity consistency. As illustrated in Figure 2, we design a **Dual-Modal Identity-Anchored Alignment** (Section 3.2) mechanism that integrates learnable identity queries and dual-modal perceptual supervision to ensure precise identity alignment; And we propose a **Sparse Routing Strategy** (Section 3.3) that adaptively assigns each video token to its most relevant subject, constraining variable-length conditioning sequences to a fixed constant size and mitigating identity entanglement. We also construct a **Multi-Subject Annotated Video Dataset** (Section 3.4) equipped with fine-grained reference annotations, addressing the data bottleneck in this field.

### 3.1. Preliminaries

DiTs (Peebles & Xie, 2023) use a 3D variational autoencoder (VAE) (Kingma & Welling, 2014) to compress videos into a latent space as $z_0$. Following the probability-flow ordinary differential equation (ODE) defined in flow matching (Lipman et al., 2023), a transformer $\epsilon_\theta$ is trained to predict the linear velocity $v_t = z_1 - z_0$ from random Gaussian noise $z_1 \sim \mathcal{N}(0, 1)$ to $z_0$ under text condition $c$. The training objective at timestep $t$ is:

$$\mathcal{L}_{\text{flow}} = \mathbb{E}_{z_0, z_1, c, t}\big[\|\epsilon_\theta(z_t, c, t) - v_t\|_2^2\big], \qquad (1)$$

$\epsilon_\theta$ is composed of stacked blocks with spatio-temporal self-attention and cross-modal cross-attention to model temporal consistency and textual control. To introduce visual conditions for $N$ reference subjects, prior works encode reference images with the same VAE, obtaining $\{c_i\}_{i=1}^N$, and concatenate them with the latent sequence $z_t$ to form

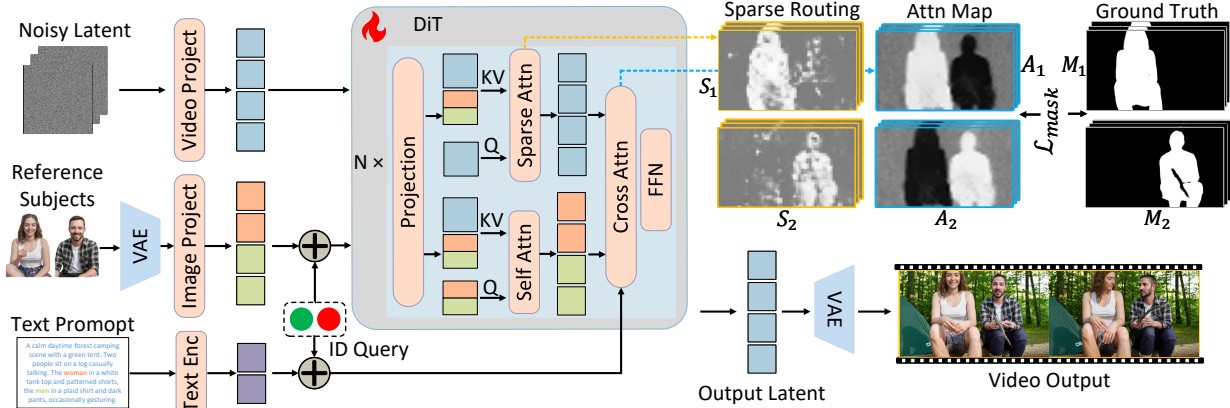

*Figure 2.* **Framework of our DiasR.** It takes noisy latents, multi-subject reference images, and text prompts as inputs, and integrates a Dual-Modal Identity-Anchored Alignment mechanism to anchor the identity of each subject, together with a Sparse Routing Strategy that mitigates the extra computational overhead induced by subject tokens.

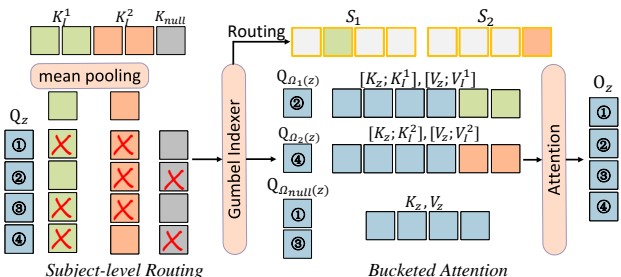

*Figure 3.* **Sparse Routing Strategy.** Left: Subject-level routing selects the most relevant subject for each query token. Right: Bucketed attention computes attention independently for each subject bucket, avoiding performance degradation while preserving cross-subject interactions.

$s_t = [z_t; \{c_i\}_{i=1}^N]$. Consistency with visual conditions is subsequently learned through self-attention over $s_t$. Because the reference images share the latent space with $z_t$, the model learns dependencies between video frames and visual conditions with minimal architectural modifications, thereby enabling efficient adaptation to subject-specific cues.

### 3.2. Dual-Modal Identity-Anchored Alignment

Existing multi-subject video generation methods typically fuse tokens from multiple reference subjects indiscriminately for in-context conditioning, which fail to disentangle identity-specific cues of each individual subject and lead to identity drift or entanglement. To address these issues, we propose a Dual-Modal Identity-Anchored Alignment Mechanism, which comprises two complementary modules: learnable identity queries for cross-modal subject association and dual-modal perceptual supervision for subject-specific semantic mapping. Collectively, we implicitly model the semantic matching between the video latent space and each subject's representation across visual and textual modalities, thereby achieving fine-grained identity consistency.

**Learnable Identity Queries.** To establish cross-modal

connection for each subject, we instantiate a set of learnable identity queries $q_{id} \in \mathbb{R}^{M \times d}$, where $M$ is the maximum number of identity queries. Each query is dedicated to tagging one distinct subject across visual and textual modal. They are not permanently bound to specific entities; instead, they provide discrete identity anchors for multiple subjects and can be viewed as a form of subject positional encoding with a fixed upper limit on the number of subjects. Specifically, we project identity query corresponding to the number of subjects into both textual and visual feature space, and add them into the subject-related text tokens and visual tokens, respectively. This design ensures that each subject's identity is explicitly anchored in both modalities, creating a cross-modal bridge that links the same subject's textual description and visual appearance. As fixed identity anchors, the learnable identity queries enable the model to precisely track and distinguish individual subjects during the generation process, effectively avoiding entanglement between multiple subjects and reinforcing the binding of each subject's cross-modal identity cues, ensuring consistent subject representation throughout the generated content.

**Dual-Modal Perceptual Supervision.** To further reinforce identity consistency at the semantic level, we supervise the perceptual ability for subject semantic mapping across both modalities. Within each transformer block, we compute dual-modal perceptions of the video sequence with each subject $i$, which are derived based on the identity queries: (1) visual similarity score $S_i \in \mathbb{R}^{b \times hw \times d}$ (as defined in Equation (4)) between the video latents and the $i$-th reference subject in self-attention, reflecting the visual perception of the subject in the video; (2) textual attention map $A_i \in \mathbb{R}^{b \times hw \times d}$ that quantifies the attention weight allocated by the video latents to the $i$-th subject's textual query in cross-attention, characterizing the textual perception of the subject. These two perceptual signals collectively capture the spatiotemporal correlation between the video

sequence and the subject's identity across modalities. We then supervise each $\mathbf{S_i}$ and $\mathbf{A_i}$ with the subject-specific ground-truth video mask $\mathbf{M_i} \in \mathbb{R}^{b \times hw \times d}$ using MSE loss:

$$\mathcal{L}_{\text{mask}} = \sum_{i=1}^{N} \left[ \|\mathbf{S}_i - \mathbf{M}_i\|_2^2 + \|\mathbf{A}_i - \mathbf{M}_i\|_2^2 \right], \quad (2)$$

This supervision enforces per-subject cross-modal semantic consistency by aligning both visual and textual perception with the subject's intrinsic spatiotemporal semantic distribution. It ensures that the dual-modal perceptions are tightly coupled with each subject's unique identity. This not only encourages the model to learn fine-grained, subject-specific semantic representations but also effectively mitigates identity inconsistency and drift in multi-subject video generation, achieving stable and consistent identity preservation.

### 3.3. Sparse Routing Strategy

The dominant computational cost for DiTs stems from the quadratic scaling of self-attention with respect to sequence length. For a latent video with $f$ frames and spatial dimensions $h \times w$, the computational complexity is approximately $\mathcal{O}\big((fhw)^2\big)$. Under in-context conditioning with $N$ reference subjects, each contributing $hw$ tokens, the sequence length increases to $(f+N)hw$, resulting in a computational complexity of $\mathcal{O}\big(h^2 w^2 (f+N)^2\big)$, which introduces an overhead of approximately $10\%$–$20\%$ in practice. He et al. (He et al., 2025) observed that the top $50\%$ of reference tokens are responsible for over $85\%$ of attention mass from noisy video latent, which indicates significant redundancy.

**Dynamic subject-level routing.** To mitigate the extra cost in multi-subject generation, we propose a subject-level *sparse reference* strategy that preserves only the most informative reference subject per query token before self-attention, as shown in Figure 3. Let the noisy video latents be $H_z \in \mathbb{R}^{b \times fhw \times d}$ and the $i$-th reference image latents be $H_I^i \in \mathbb{R}^{b \times hw \times d}$. We project them into $Q_z, K_z, V_z$ and $K_I^i, V_I^i$, respectively. Before full-sequence attention, we pre-estimate a similarity score $S \in \mathbb{R}^{b \times fhw \times (N+1)}$ between each query token $q$ in $Q_z$ and all subjects:

$$S(q) = \text{Softmax}\left( \frac{q\,[\phi(K_I^1); \dots ; \phi(K_I^N), K_{null}]^\top}{\sqrt{d}} \right), \quad (3)$$

where $[\cdot]$ denotes concatenation and $K_{null} \in \mathbb{R}^{b \times f \times d}$ is an additional empty token to gather those tokens that are unrelated to all subjects (mostly the background). We introduce a descriptor $\phi(\cdot)$ over spatial tokens (mean pooling over $hw$). Previous work like CLIP (Radford et al., 2021) have demonstrate that this efficient operation can capture the dominant semantic features with simple representational

capabilities. Although mean pooling discards some details, it is sufficient for similarity matching. The visual similarity score $S_i$ is obtained by aggregating the scores of $S(q)$ for the reference subject $i$ at each token $q$ in $Q_z$:

$$S_i : \quad S_i(q) = S(q)_i, \ 0 \le i < N, q \in Q_z. \quad (4)$$

**Subject-Sparse Bucketed Attention.** Since each token $q$ in the video sequence typically depends on a single subject, we select the top-1 subject index using the Gumbel-Softmax (Jang et al., 2017) for differentiability:

$$Ind(q) = \arg\max(\text{Gumbel}(S(q))), \quad (5)$$

and apply these index to the self-attention over the in-context conditioning sequence. The top-1 routing works at the subject-level rather than the token-level, each token in the video sequence interacts only with its most relevant subject. Extending to top-k routing would allow each video token to interact with multiple subjects simultaneously, increasing the risk of identity entanglement and feature blending. Therefore, we adopt strict top-1 routing to ensure one-to-one token-subject correspondence.

Based on the per-token subject index $Ind(q)$, we can straightforwardly construct a naive attention mask to filter out irrelevant subjects. However, attention acceleration kernels such as FlashAttention (Dao et al., 2022) typically do not support arbitrary sparse attention masking patterns, which results in significant degradation in practical runtime performance. To address this limitation, we have designed a bucket-based sparse attention mechanism. We group queries into buckets $\Omega_i(z)$ if their $Ind(z) = i$ and compute attention independently within each respective bucket:

$$O_{\Omega_i(z)} = \text{Attn}\big(Q_{\Omega_i(z)}, [K_z; K_I^i], [V_z; V_I^i]\big). \quad (6)$$

This subject-level bucketed attention effectively avoids the performance penalties introduced by irregular sparse masks, while assigning each query token to a single subject to prevent spatial-temporal entanglement among multiple subjects. Meanwhile, interactions between different subjects are preserved during self-attention.

### 3.4. Multi-Subject Annotated Video Dataset

Personalized multi-subject video generation requires high-quality data with clearly annotated subjects, especially for human-centric interaction scenarios where realistic motion and pose modeling is fundamental (Liu et al., 2022). However, existing public datasets like HOIGen-1M (Liu et al., 2025a) are not explicitly designed for subject-level disentangled annotations (i.e., reference images and masks), which has become a major bottleneck. We therefore develop a multi-subject-centric, multimodal processing pipeline to

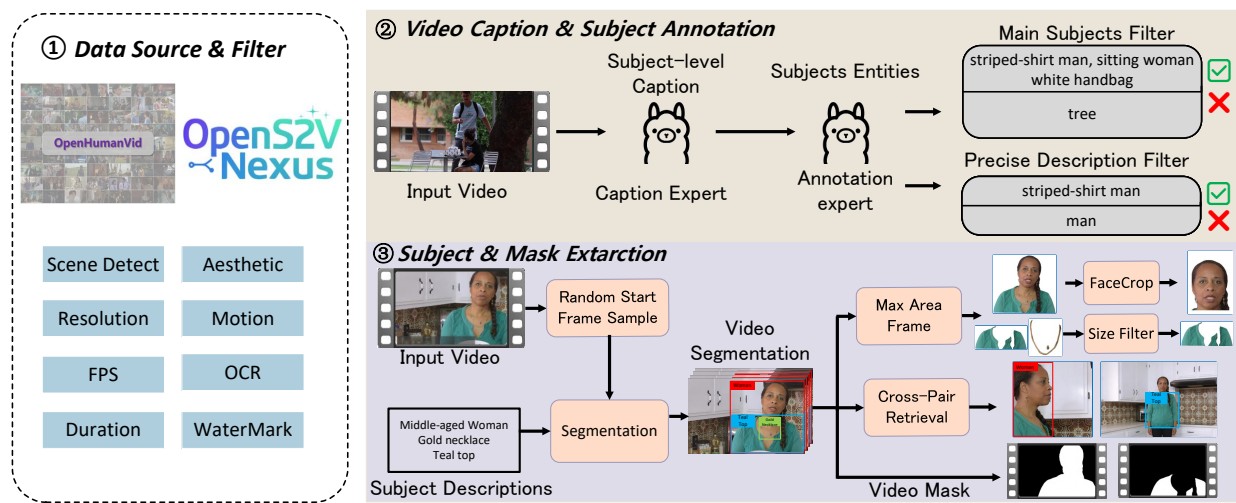

*Figure 4.* Dataset construction pipeline of **MuSA-2M**.

construct fully annotated video–text–subject–mask quadruples, as illustrated in Figure 4.

Our raw data sources include OpenHumanVid (Li et al., 2025), OpenS2V-5M (Yuan et al., 2026), and additional in-house e-commerce video clips. These datasets include numerous multi-subject scenarios containing combinations of humans and objects. We utilize vision-language models (e.g., MIMO (Xiaomi, 2025) and Intern-VL (Zhu et al., 2025)) to generate captions and dense phrases for the filtered videos, from which per-subject appearance descriptions are extracted. Based on these phrases, Grounding-SAM2 (Ren et al., 2024) is subsequently employed to obtain subject-specific reference images and pixel-level masks across the entire video. Following the extraction of subject-specific reference images associated with each video, we perform cross-video subject similarity matching by retrieving video clips featuring the same subject by comparing subject feature similarities across clips within the same video, and utilize these retrieved clips to replace the corresponding subject segments in the original video. This process ensures that the same subject within the dataset is depicted in diverse video contexts, and avoids unnatural copy-paste artifacts. To further enhance facial image quality, we additionally extract cropped facial images for all human subjects using facecrop-plus (Birškus, 2023). In total, our dataset comprises approximately 2 million video quadruples, providing a comprehensive foundation for training models capable of highly consistent multi-subject video generation. Further details are provided in Section C of the supplementary material.

## 4. Experiments

### 4.1. Experimental Settings

**Implement Details.** Our model is initialized from Wan-2.1-14B-I2V-480P (Wan et al., 2025), and the text encoder

adopts T5 (Raffel et al., 2020). For Gumbel-Softmax, we adopt the soft mode during training to preserve gradient flow, and the hard mode during inference. All training videos are uniformly sampled to 81 frames with a resolution of 480×832. We adopt two-stage training with a total of 12,000 optimization steps on 64 NVIDIA A800 GPUs using our custom dataset. The training objective comprises $\mathcal{L}_{flow}$ and $\mathcal{L}_{mask}$ terms, with weighting coefficients set to 1.0 and 0.05, respectively. At inference, we employ an Euler sampler with 50 denoising steps and classifier-free guidance (Ho & Salimans, 2022) with a guidance scale of 5. Additional implementation details are provided in Section B.2 of the supplementary material.

**Evaluation Metrics.** We adopt the OpenS2V-Eval benchmark (Yuan et al., 2026) to comprehensively evaluate the model's performance. Specifically, we evaluate 180 test cases covering multi-person/object, single-person/object, person-object interaction, and face scenarios. For each subset, we compute metrics for aesthetics, motion quality (MotionSmoothness and MotionAmplitude), facial similarity (FaceSim), text fidelity (GmeScore), multi-subject consistency (NexusScore), and video naturalness (NaturalScore), and compute their weighted sum to obtain an overall TotalScore. We adopt the official implementation available on their GitHub repository, and the metric composition differs from that presented in their paper version. The NaturalScore reported in Table 1 differs from that in the official OpenS2V-Eval release, likely due to a change in the GPT model used for evaluation. Additional details on the metrics are provided in Section B.3 of the supplementary material.

**Baseline.** We compare our method with recent open-source multi-subject video generation models on both the OpenS2V-Eval benchmark and our custom multi-subject-text dataset, performing both qualitative and quantitative evaluations. The baseline models include HuMo (Chen

| Method | TotalScore ↑ | Aesthetics ↑ | Motion Smoothness ↑ | Motion Amplitude ↑ | FaceSim ↑ | GmeScore ↑ | NexusScore ↑ | NaturalScore ↑ |
|---|---|---|---|---|---|---|---|---|
| Pika-2.1 (Labs, 2025) | 49.38% | 46.88% | 87.06% | 24.71% | 30.38% | 69.19% | 45.40% | 63.32% |
| Vidu-2.0 (AI, 2025b) | 48.87% | 41.48% | 90.45% | 13.52% | 35.11% | 67.57% | 43.37% | 65.88% |
| kling-1.6 (AI, 2025a) | 56.32% | 44.59% | 86.93% | **41.60%** | 40.1% | 66.2% | 45.89% | **82.52%** |
| Phantom-1.3B (Liu et al., 2025b) | 53.78% | 46.67% | 93.30% | 14.29% | 48.56% | 69.43% | 42.48% | 71.06% |
| Phantom-14B (Liu et al., 2025b) | 55.38% | 46.39% | 96.31% | 33.42% | 51.46% | 70.65% | 37.43% | 74.03% |
| VACE-1.3B (Jiang et al., 2025) | 48.67% | 48.24% | 97.20% | 18.83% | 20.57% | **71.26%** | 37.91% | 73.47% |
| VACE-14B (Jiang et al., 2025) | 55.96% | 47.21% | 94.97% | 15.02% | 55.09% | 67.27% | 44.08% | 73.75% |
| HuMo-1.7B (Chen et al., 2025) | 50.49% | 38.54% | 95.64% | 13.23% | 57.53% | 68.56% | 42.16% | 56.06% |
| HuMo-17B (Chen et al., 2025) | 56.22% | 48.39% | **97.97%** | 20.10% | 55.37% | 66.19% | 41.29% | 75.15% |
| Kaleido-14B (Zhang et al., 2025b) | 55.83% | **48.66%** | 97.57% | 13.40% | 47.97% | 69.24% | 41.09% | 79.86% |
| Ours | **57.79%** | 45.87% | 97.47% | 16.48% | **60.66%** | 66.60% | **47.40%** | 74.60% |

*Table 1.* **Quantitative comparison with other methods.** The 1st and 2nd ranks for each metric are marked in **bold** and underlined.

et al., 2025), VACE (Jiang et al., 2025), Phantom (Liu et al., 2025b), and Kaleido (Zhang et al., 2025b). We also report the official OpenS2V-Eval metrics for several closed-source systems—Kling 1.6 (AI, 2025a), Vidu 2.0 (AI, 2025b), and Pika 2.1 (Labs, 2025)—to contextualize our results within the broader research landscape. The metrics of all baselines strictly follow the official standard settings of OpenS2V-Eval. We fully adopt the recommended settings of all baselines. OpenS2V-Eval only provides reference subjects, prompts, and metric computation.

## 4.2. Comparison

With the exception of Kaleido, each open-source baseline has two variants with different model sizes. In our quantitative evaluation, we report metrics for all variants; for the qualitative evaluation, we only use the larger variant of each method to ensure better visual quality.

**Quantitative Comparison.** We evaluate all methods on the OpenS2V-Eval benchmark. The generated videos of HuMo and Kaleido are evaluated under their official protocols, while other videos are taken from the evaluations released by the OpenS2V team. As shown in Table 1, our method achieves the best performance on both FaceSim and NexusScore—the two key metrics for evaluating identity consistency—and this superiority is attributed to our accurate cross-modal identity alignment and sparse routing strategy, which suppresses interference from irrelevant subjects. On other general metrics, our method is comparable to state-of-the-art approaches. Although constrained by the dataset scale and cross-domain diversity, our method is inferior to certain bias-mitigating methods trained on large-scale real-world datasets in terms of video naturalness, yet it still achieves optimal overall performance.

**Qualitative Comparison.** We evaluate our approach across representative scenarios (human–object interaction, multi-human interaction) against competitive baselines. Beyond the core requirement of multi-identity consistency, the multi-human and human–object settings primarily probe bidirectional dynamic interaction modeling. As shown in Figure 5, other methods often exhibit identity drift, omissions, or en-

tanglement: VACE frequently loses identity consistency in multi-subject scenes; Phantom displays identity entanglement; HuMo sometimes mishandles multi-person scenes, leading to duplicated or entangled subjects. Kaleido also occasionally suffers from identity inconsistency and drift. In contrast, our method consistently preserves the identities of multiple subjects and correctly models interactions across different subject groups, owing to our subject-level identity alignment and sparse subject routing mechanism.

**Computational Performance Comparison.** To demonstrate the effectiveness of the proposed sparse reference strategy, we present Figure 7, which plots the one-step inference time versus the number of reference entities, and the one-step floating-point operations (FLOPs) in the self-attention layer. HuMo shares the same self-attention layer as Phantom, but its extra audio block results in a longer runtime. Owing to our Sparse Routing Strategy with a constant token budget, our method exhibits nearly constant inference time as the number of reference entities increases and minimal growth in FLOPs, while other baseline approaches grow rapidly in runtime and FLOPs. This property substantially reduces computational costs.

## 4.3. Ablation & Analysis

To further evaluate the effectiveness of different components in our approach for multi-subject identity consistency and efficient generation, we perform an ablation study on the *Identity Query*, $\mathcal{L}_{\text{mask}}$, and the *sparse reference strategy*, respectively. All experiments are performed on our DiasR framework with identical training and evaluation protocols. Unless noted otherwise, we only remove or revise the target component for each ablation. We report the qualitative results in Figure 6 and quantitative metrics in Table 2.

**Effect of Identity Query.** We ablated the *Identity Query*, which is designed to strengthen cross-modal identity association and anchor among different subjects. Removing this component resulted in significant drops in *GmeScore*, *FaceSim*, and *NexusScore*. Qualitatively, some subjects exhibited actions deviating from the prompt, with degraded identity consistency. These results indicate that the *Identity*

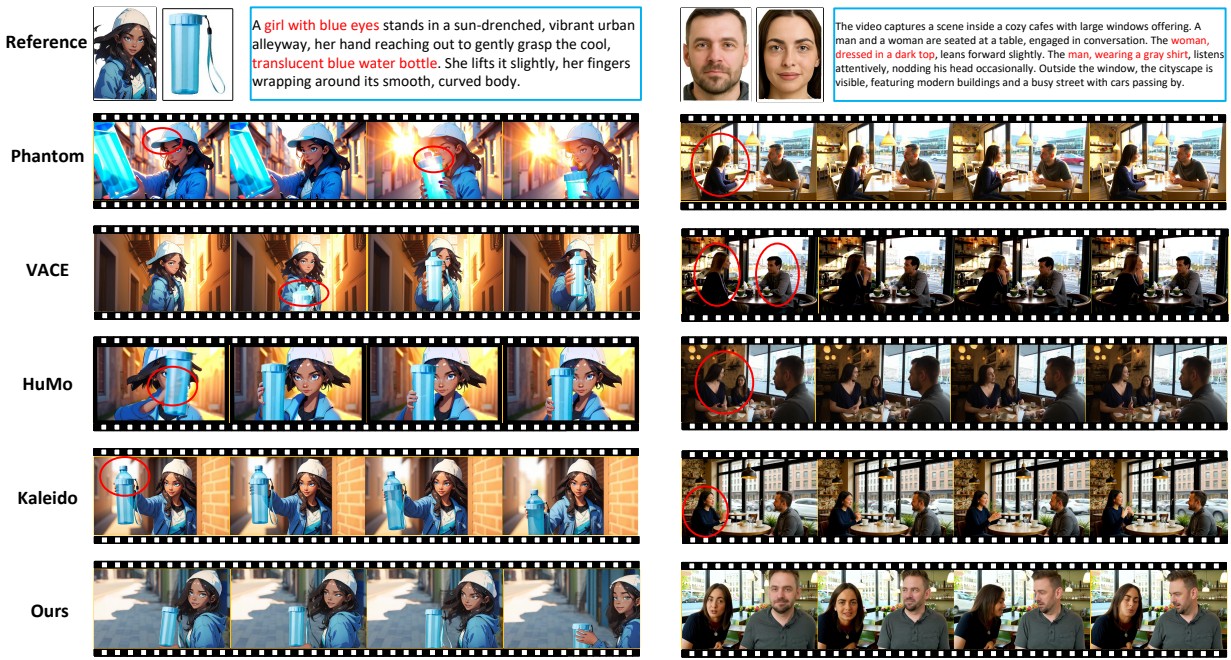

*Figure 5.* **Qualitative Comparison against other methods.** The red circles mark some failed details.

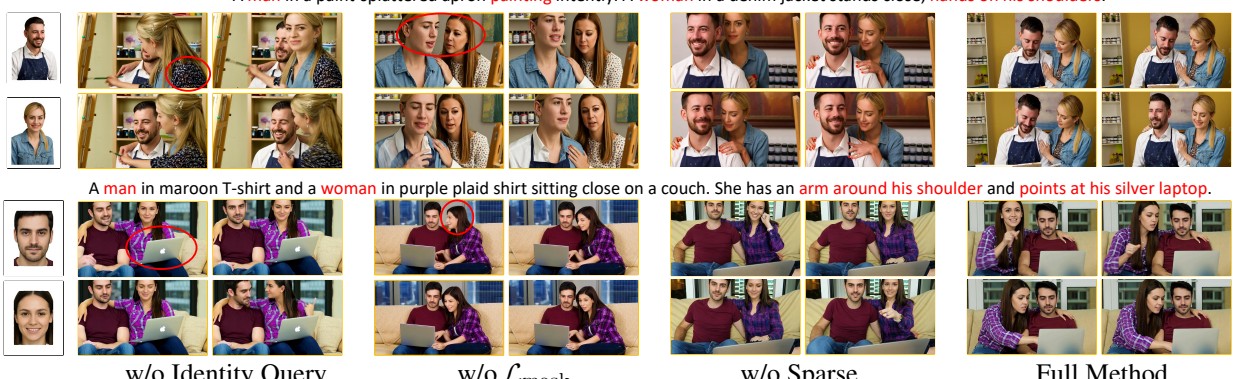

*Figure 6.* **Ablation Study.** We ablate variants w/o Identity Query, $\mathcal{L}_{mask}$, or the sparse reference strategy, against our full method.

*Query* is crucial for aligning dual-modal representations, as well as preserving identity across multiple subjects.

**Effect of $\mathcal{L}_{\text{mask}}$.** The $\mathcal{L}_{\text{mask}}$ loss is designed to supervise subject-level dual-modal alignment. When $\mathcal{L}_{\text{mask}}$ is removed, the consistency metrics (i.e., *FaceSim* and *NexusScore*) decrease significantly, and qualitative results reveal identity entanglement between subjects and identity drift. This demonstrates that $\mathcal{L}_{\text{mask}}$ is essential for learning disentangled, subject-specific dual-modal representations in multi-subject settings.

**Effect of Sparse Routing Strategy.** Because visual supervision arises from descriptors in subject-level routing, we ablate only the bucketed attention and allow the model to perform in-context conditioning over the full token sequence. As shown in Table 2, Our *Sparse Routing Strategy*

achieves quantitative and qualitative performance comparable to that of full-sequence attention, with the only exception of MotionAmplitude. The reason is the top-1 routing eliminates chaotic motion from irrelevant subjects while preserving temporal consistency. The slightly increased NaturalScore proves that it does not compromise video realism. We also achieve relatively faster runtime as the token sequence length increases, as shown in Figure 7. This demonstrates that in multi-subject scenarios, our approach can efficiently generate videos while maintaining high visual quality and identity consistency.

### 4.4. Human Preference Study

To fully capture the perceptual quality and interaction realism of multi-reference video generation, we further conduct

| Method | TotalScore ↑ | Aesthetics ↑ | Motion Smoothness ↑ | Motion Amplitude ↑ | FaceSim ↑ | GmeScore ↑ | NexusScore ↑ | NaturalScore ↑ |
|---|---|---|---|---|---|---|---|---|
| w/o ID Query | 52.33% | 46.67% | 93.30% | 14.29% | 48.56% | 64.47% | 42.48% | 67.50% |
| w/o $\mathcal{L}_{mask}$ | 49.54% | 45.02% | 93.17% | 21.81% | 30.83% | **69.43%** | 43.04% | 66.90% |
| w/o Sparse | 57.67% | 44.39% | 96.31% | **33.42%** | 59.46% | 66.65% | 46.93% | 72.35% |
| Full Method | **57.79%** | 45.87% | 97.47% | 16.48% | **60.66%** | 66.60% | **47.40%** | **74.60%** |

*Table 2.* **Ablation Study.** The 1st and 2nd ranks for each metric are marked in **bold** and underlined.

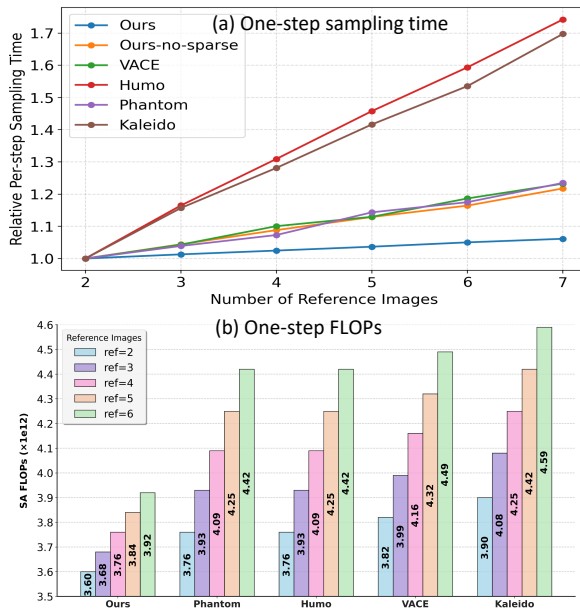

*Figure 7.* **One-step Computational Comparison.** We evaluate computational cost through one-step sampling time in (a) and floating-point operations (FLOPs) in (b).

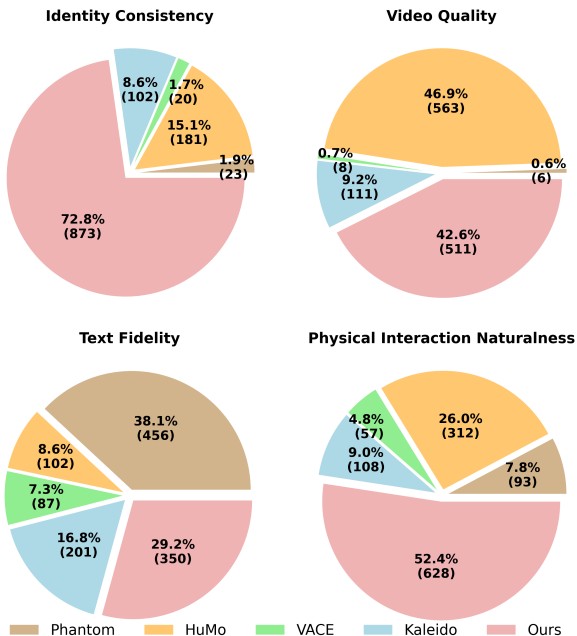

*Figure 8.* **Human Preference Study.** 30 participants selected the best-performing method from five methods including ours across 40 test cases, with evaluations on four core criteria.

a human preference study to comprehensively evaluate our method against other baselines. We recruited 30 participants and evaluated four core dimensions of multi-reference video generation: Identity Consistency, Video Quality, Text Fidelity, and Interaction Naturalness. The study is conduct on 40 cases from our test set under a forced-choice evaluation protocol, where participants selected the single best-performing method for each case across all four dimensions. We then counted the total number of times each method was chosen as the most preferred.

Quantitative results in Figure 8 demonstrate that our approach achieves dominant performance on the most critical dimensions. Specifically, our method is selected as the most preferred approach for Identity Consistency (72.75%) and Interaction Naturalness (52.42%) by a substantial margin, significantly outperforming all baselines. For Video Quality and Text Fidelity, our method remains the second-best performer, achieving competitive preference rates of 42.58% and 29.25%, respectively. Overall, the user study validates the superior effectiveness of our method for multi-reference video generation, particularly in preserving identity consistency and modeling natural physical interactions.

## 5. Conclusion

In this paper, we present DiasR, an efficient identity-consistent multi-subject video generation approach to address the cross-frame identity drift and prohibitive computational costs in personalized multi-subject video generation. DiasR integrates Dual-Modal Identity-Anchored Alignment, a Sparse Routing Strategy, and the novel MuSA-2M dataset. MuSA-2M fills the gap in subject-level annotated multi-subject video data, providing a robust foundation for model training. The Dual-Modal Identity-Anchored Alignment enforces precise identity consistency across visual and textual modalities through supervised alignment with ground-truth masks, effectively mitigating identity drift. The Sparse Routing Strategy curbs quadratic computational growth via dynamic subject-level routing and bucket aggregation, ensuring efficiency without compromising interaction fidelity. Experiments demonstrate that DiasR outperforms baselines in identity consistency, text fidelity, and inference efficiency, maintaining stable runtime as the number of reference subjects increases. This work enables high-quality, controllable content creation and advances the practical applicability of multi-subject video generation in real-world scenarios.

## Acknowledgements

This work was supported by the National Key Research and Development Program of China under Grant 2024YFE0203200, the National Nature Science Foundation of China under Grant U24A20329 and Grant 62527810, the Fundamental and Interdisciplinary Disciplines Breakthrough Plan of the Ministry of Education of China under Grant JYB2025XDXM103, and the Science Fund for Creative Research Groups under Grant 62121002.

## Impact Statement

The proposed **DiasR** enables identity-consistent multi-subject video generation via efficient sparse routing and identity-aware dual alignment, democratizing high-quality content creation by lowering computational barriers for creators and non-profits. It supports reliable multi-subject interactions in education, healthcare, and cultural preservation, addressing field-wide data gaps and setting a benchmark for efficient, controllable video generation while fostering creative diversity. Notably, it carries risks: realistic deepfakes information may erode public trust, and unauthorized use of copyrighted material raises legal/ethical concerns. To mitigate these, user-facing applications must implement robust security reviews for harmful content, mandatory transparent labeling of AI-generated outputs, and rigorous pre-screening of training data to ensure copyright compliance, thereby balancing innovation with responsible use.

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

# A. Overview

In the supplementary materials, we introduce more detailed analysis and additional results as follows:

- Implementation details ( Section B):
    - Training phrase ( Section B.1);
    - Model details ( Section B.2);
    - Evaluation metrics ( Section B.3);
    - Test dataset ( Section B.4);

- Curation of Multi-Subject Annotated Video Dataset ( Section C):

- Additional experiments ( Section D):
    - More comparison results ( Section D.1);
    - More visualization results ( Section D.2);

- Limitations ( Section E):

# B. Implementation Details

## B.1. Training Phase

Our model is initialized with Wan2.1-14B-i2v-480p, which can generate videos with an image as the first frame. Leveraging this inherent advantage, our primary training objectives are to endow the model with reasonable customization capabilities in multi-subject scenarios and to avoid mere copy-pasting of reference images. To ensure the model generates high-quality videos with consistent subject identities, the training process is divided into two stages to progressively enhance the model's performance.

Stage 1 is dedicated to cultivating the model's multi-subject customization capacity and modeling rational interactions between different subjects. We use multi-subject data from the MuSA-2M dataset as input. The core aim here is to initially model reasonable transformations from reference images to videos and capture the interactive dynamics among multiple subjects. This stage contains 10,000 steps. Stage 2 focuses on higher identity consistency and video quality. We perform strict data filtering on the MuSA-2M dataset and select 100,000 high-quality samples based on aesthetic scores, technical scores, etc. These carefully curated samples enable the model to learn refined visual features and temporal consistency, thereby further elevating the quality of generated videos. This stage contains 2,000 steps.

Our training is conducted on 64 Nvidia A800 GPUs. Throughout the training process, the resolution is uniformly set to 480 × 832. This fixed resolution configuration ensures consistency in training inputs and helps the model concentrate on optimizing subject consistency capabilities.

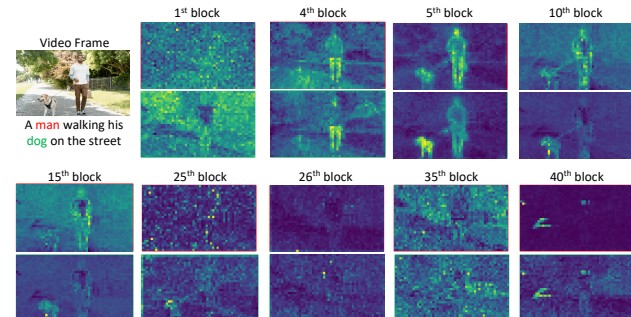

*Figure 9.* **Cross-attention visualization of DiT's blocks.** The 5th to 25th block have richer semantic.

| Blocks | 1~5 | 6~10 | 11~15 | 16~20 |
|---|---|---|---|---|
| MSE | 0.216 | 0.084 | 0.112 | 0.093 |
| Blocks | 21~25 | 26~30 | 31~35 | 36~40 |
| MSE | 0.098 | 0.121 | 0.146 | 0.175 |

*Table 3.* MSE Values Across Different Blocks

## B.2. Model Details

We initialize our model from Wan-2.1-14B-I2V-480p (Wan et al., 2025), which adopts *patch_embedding* as the projection layer to map video latent features into the self-attention space. To better differentiate between video features and reference image features in the in-context token sequence, we introduce a learnable *ref_embedding* (with the same dimension as *patch_embedding*) to project reference image features. This addresses the temporal discontinuity along the frame dimension caused by the direct concatenation of reference image latent features and video latent features.

For textual modal perceptual supervision, the certain blocks of DiT capture richer semantic details. To further quantitatively evaluate the perceptual capability of different blocks toward the text modality, we employ the original Wan2.1-I2V-14B-480p model to generate videos from 100 reference images with one subject. The prompt was uniformly set to: a [object Object] in [object Object]. For each generated video, we extract the attention map of each DiT block and average them across 50 timesteps. We then compute the mean squared error between the normalized attention map and the ground-truth mask of the corresponding subject. As shown in Table 3, blocks 6 to 25 achieve the optimal textual perception capability. We also visualize the attention maps of 40 blocks in Wan2.1-14B-I2V model for subject words in the prompt, as shown in Figure 9, the results indicate that the block 5th to 25th exhibit clearer attention responses to prompt.

Base on this observation, we collect attention maps $A_i$ from these representative blocks (5th ~25th blocks). For visual modal perceptual supervision, we implement supervi-

sion across all blocks to ensure the sparsity of the network. The empty token $K_{null}$ is designed as an 1-dimensional learnable embedding matching the query dimension in self-attention, while the identity queries consist of $M$ learnable embeddings with the same dimensions as $K_{null}$ (where $M$ is preset to 10). During training, we randomly sample N queries that match the number of reference subjects for identity-aware optimization.

## B.3. Evaluation Metrics

Our evaluation metric is derived from OpenS2V-Nexus (Yuan et al., 2026), which focuses on assessing the model's capability to generate videos with consistent subjects and ensures the natural appearance and identity consistency of the subjects. It introduces eight primary categories for subject-to-video generation, including multi-human, multi-face, multi-object, single-face, single-human, single-object, human-object and face-object, comprising a total of 180 distinct test samples. The generated videos are then evaluated against the metric set $\mathcal{M}$. We implement all metrics based on its official GitHub repository.

**Aesthetics:** This metric is designed to evaluate the aesthetic quality of videos. It adopts the aesthetic predictor (Schuhmann et al., 2022), which involves feeding video frames directly into the model to generate aesthetic scores; the mean of all valid scores is then calculated to yield the aesthetic score for the video.

**Motion Smoothness:** This metric is dedicated to evaluate the motion smoothness of generated videos, and it leverages a multimodal model MPLUG-Owl2 (Ye et al., 2024) to comprehensively assess the inter-frame motion coherence and transition fluency of video sequences, thereby quantifying the overall motion smoothness performance of the video.

**Motion Amplitude:** This metric is designed to assess the motion magnitude of generated videos, which employs the OpenCV (Bradski & Kaehler, 2000) to calculate by OpticalFlowFarneback (Farnebäck, 2003). This approach computes the dense optical flow fields between consecutive video frames, and further quantifies the overall motion magnitude of the entire video frames based on the flow field data.

**FaceSim:** This metric is dedicated to evaluate the consistency between facial regions in videos and reference images. It adopts ConsisID [123], which first leverages the InsightFace toolkit [17] to accurately detect and localize facial regions in both video frames and reference images; second, the similarity between the detected facial regions is computed in CurricularFace feature space [34]; finally, all valid frame-level similarity scores are aggregated through summation and then arithmetically averaged to derive the final facial consistency score for the entire video sequence.

**GmeScore:** This metric is used to evaluate the consistency between videos and text prompts. It employs GME (Zhang et al., 2024), a model fine-tuned on Qwen2-VL (Wang et al., 2024), which calculates the similarity between text embeddings and video frame embeddings. This model inherently supports text prompts of variable lengths and is capable of producing more reliable relevance scores, making it well-suited for the long-prompt scenarios in DiT-based video generation tasks.

**NexusScore:** The NexusScore evaluates the subject consistency between video frames $\{V_t\}_{t=0}^{T}$ and reference images $\{I_i\}_{i=0}^{N}$, which leverages an image-text detection model $M_{\text{detect}}$ (Cheng et al., 2024) and a multimodal retrieval model $M_{\text{retrieve}}$ (Zhang et al., 2024). Specifically, the reference images $I_i$ and video frames $V_t$ are fed into $M_{\text{detect}}$ to generate the corresponding bounding boxes of these targets and perform cropping. Then, the similarity between the cropped entity regions $E_{i,t}$ and the target subject names $T_{i,t}$ is calculated by $M_{\text{retrieve}}$ as $s_{i,t} = M_{\text{retrieve}}(E_{i,t}, T_{i,t})$. If both the confidence score of the bounding box and $s_{i,t}$ exceed the predefined thresholds, the NexusScore can be calculated as: NexusScore $= \frac{1}{N \times T'} \sum_{i=0}^{N} \sum_{t=0}^{T'} M_{\text{retrieve}}(E_{i,t}, I_i)$ where $T'$ is the number of frames in which a subject is detected.

**NaturalScore:** The NaturalScore is calculated by simulating human evaluators with GPT-4o (Achiam et al., 2023) to evaluated the naturalness of the generated videos (i.e., their compliance with physical laws). Specifically, a 5-point evaluation criterion is designed based on common sense and physical laws, denoted as $\mathcal{C} = \{c_1, c_2, c_3, c_4, c_5\}$, where each $c_i$ stands for the score corresponding to a specific evaluation level. The video frames are fed into GPT-4o and get a score in accordance with the 5-point criterion. The final score $S_{\text{Natural}}$ is calculated as the average of the sampleds frames.

All video frames are utilized for the MotionAmplitude metric, while other metrics uniformly sample 32 frames for computations. After completing the primary calculation of each individual metric, Every metrics are subjected to linear normalization within a predefined value range, and then averaged across the evaluation set. The TotalScore is formulated as a weighted linear combination of all metrics as: $TotalScore = \sum_{i \in \mathcal{M}} w_i \cdot S_i$, where $S_i$ represents the score of the $i$-th metric, and the weight $w_i$ for each metric is assigned as follows: 0.20 for NexusScore, 0.24 for NaturalScore, 0.12 for GmeScore, 0.20 for FaceSim-Cur, 0.16 for AestheticScore, 0.06 for MotionSmoothness, and 0.02 for MotionAmplitude.

## B.4. Test Dataset

We conducted extensive qualitative and quantitative evaluation experiments on the OpenS2V (Yuan et al., 2026) dataset. To further evaluate the video quality and identity consistency of the proposed method, we curated 44 human images and 24 object images, which were randomly paired two at a time to form a total of 40 image pairs covering three categories: multi-human, multi-object, and human-object. For each pair, we employed QWen2.5-VL (Bai et al., 2025) to generate tailored text prompts for the corresponding subjects. Additionally, we pre-segmented all subjects onto a plain white background by SAM (Kirillov et al., 2023). Several examples from the constructed test set are illustrated in Figure 10.

## C. Multi-subject Annotated Video Dataset

Our Multi-subject annotated video dataset (MuSA-2M) is constructed to address the bottleneck of insufficient subject-level paired annotations in existing public video datasets, providing high-quality, multimodal quadruples (video-text-subject-mask) for training consistent multi-subject video generation models. As shown in Figure 4, the dataset pipeline consists of three core stages: (1) Data Source and Filtering, (2) Video Captioning and Subject Annotation, (3) Subject and Mask Extraction, which systematically transforms raw video data into a fully annotated dataset. We indicate in Table 4 the total number of videos retained from different sources at each stage.

All thresholds are solely for data cleaning and are not learnable hyperparameters for the model. Theoretically, higher filtering thresholds result in better data quality. Data quality and quantity exhibit a unimodal trend. Therefore, a reasonable trade-off between data quality and quantity is sufficient. Table 5 shows the percentage of clean data at some thresholds that are not mandatory, demonstrates the effectiveness of our threshold selection.. We conducted manual sampling after cleaning to ensure dataset reliability.

### C.1. Data Source and Filter

**Data Source.** The raw video data is curated from three heterogeneous sources to ensure diverse multi-subject scenarios:

- **OpenHumanVid (Li et al., 2025):** A large-scale human-centric video dataset with rich daily-life interactions, providing diverse multi-human scenes.

- **OpenS2V Nexus (Yuan et al., 2026):** A multimodal video dataset with 5M+ clips, covering multi-object interactions and human-object interaction scenarios.

- **In-house E-commerce Videos:** Proprietary video data focusing on product-human interactions, supplementing domain-specific multi-subject cases.

These sources collectively cover multi-human, multi-object, and human-object interaction scenarios, laying a foundation for multi-subject video generation.

**Filter Pipeline.** To ensure data quality, we apply a multi-dimensional filtering pipeline to preprocess raw videos:

- **Scene Detection:** Filter out videos that contain scene changes. to retain logically coherent visual content.

- **Aesthetic Filtering:** Retains videos with high visual quality (aesthetic score $> 5.8$), balanced composition, and natural lighting, avoiding low-aesthetic footage that harms annotation reliability.

- **Resolution & FPS Filtering:** Enforces a minimum resolution threshold as 480p and stable frame rate (24–30 FPS) to guarantee temporal consistency and clarity.

- **Motion & Duration Filtering:** Selects videos with moderate motion (avoiding static frames or extreme blur) and optimal duration (5–10 seconds) to balance content richness and processing efficiency.

- **OCR & Watermark Filtering:** Removes videos with interfering text or visible watermarks, ensuring data cleanliness.

This filtering stage retaining high-quality samples suitable for subsequent annotation.

### C.2. Video Caption and Subject Annotation

This stage involves generating subject-level captions for the videos and extracting subject entity words from them.

**Subject-level Caption and Entity Generation.** We utilize state-of-the-art vision-language models, including MIMO (Xiaomi, 2025) and Intern-VL-3 (Zhu et al., 2025), as caption experts and annotation experts, respectively, to generate video-level captions and subject-level dense descriptions. First, the caption expert analyzes the video, generating detailed descriptions that include multiple subject entities and the interaction relationships between them. Subsequently, the annotation expert extracts discrete subject entities and their corresponding descriptions from these captions.

**Description Filter.** Descriptions of subject entities must be meaningful and sufficiently discriminative to provide unique clues for subsequent segmentation. Therefore, we filter out certain subject descriptors from two perspectives: (1) non-essential subjects (e.g., trees in the background); (2) overly simplistic descriptors (e.g.,"striped-shirt man" better than

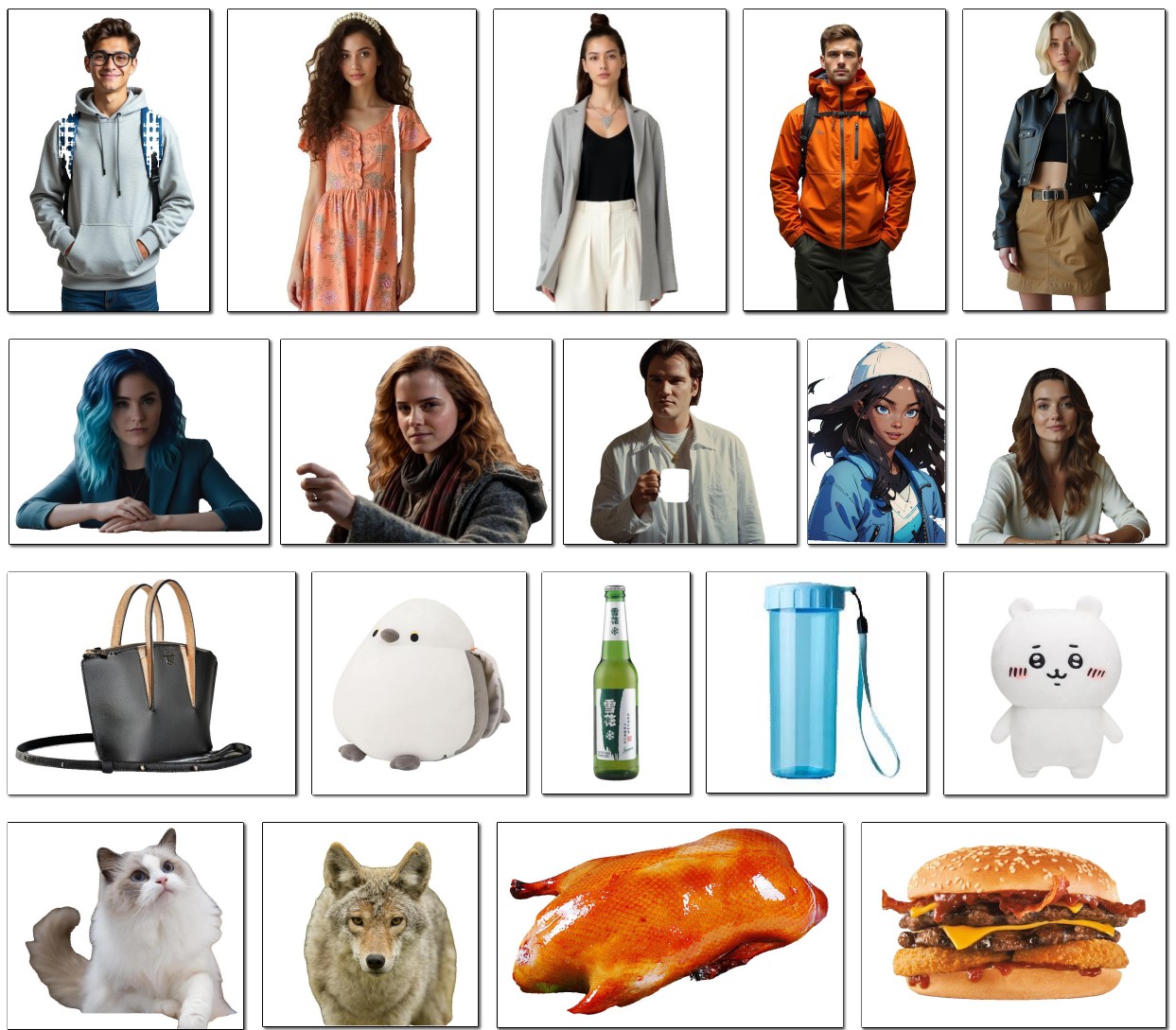

*Figure 10.* **The collated test dataset.** We curate 40 multi-subject examples encompassing diverse combinations of humans, animals, products, foods, and other object categories.

"man"). After this filtering, we exclude videos with fewer than two subjects based on the subject-level descriptions.

Following this stage, we obtained videos paired with precise, filtered captions and subject entity descriptions, enabling accurate segmentation.

### C.3. Subject and Mask Extraction

This stage generates pixel-level masks and high-quality reference images for each annotated subject, forming the final video-text-subject-mask quadruples:

**Video Frame Segmentation.** We use Grounded-SAM2 (Ren et al., 2024) for video segmentation. Since this model requires selecting an anchor frame for initialization and some subjects may not appear in every frame, we randomly sample starting frames from each video to balance computational efficiency and temporal coverage. This ensures representative frames are selected for segmentation without processing every frame. We feed the subject descriptions as input to Grounded-SAM2, which performs pixel-level segmentation on the sampled frames and full video sequences, producing per-frame subject masks. For each subject, we identify the frame where it occupies the largest pixel area, apply the mask to obtain a white-background image of the subject, and use this as a reference image containing the clearest details of the subject. For human subjects, we further apply FaceCrop (Birškus, 2023) to extract facial regions and filter out overly small objects (screen occupancy $< 0.1$ for humans and $< 0.04$ for other subjects) based on size, ensuring the extracted reference images meet resolution requirements.

| Dataset Source | Data Filter | Caption & Annotation | Subject & Mask Extraction | Cross-Paired Retrieval |
|---|---|---|---|---|
| OpenHumanVid | 10,581,543 | 7,264,215 | 1,078,659 | 610,327 |
| OpenS2V-5M | 1,041,031 | 827,362 | 583,378 | 397,014 |
| E-commerce | 1,453,035 | 917,931 | 356,952 | 132,630 |
| Total | 13,075,609 | 9,009,508 | 2,018,989 | 782,658 |

*Table 4.* **Data Volume at Different Stages.** The final dataset is from the third phase, and cross-paired retrieval enhances the high-level semantics of reference images to tackle the subject copy-paste issue.

*Table 5.* Analysis of Thresholds and Ratios in MuSA-2M

| | Threshold ($>$) | Ratio (%) | Threshold (Ours) ($>$) | Ratio (%) | Threshold ($>$) | Ratio (%) |
|---|---|---|---|---|---|---|
| Aesthetic Score | 5.0 | 79.1 | 5.8 | 32.3 | 6.0 | 9.4 |
| Motion Score | 0.005 | 81.2 | 0.01 | 61.2 | 0.05 | 9.3 |
| Human Occupancy | 0.04 | 71.5 | 0.1 | 52.3 | 0.4 | 13.2 |
| Object Occupancy | 0.01 | 63.2 | 0.04 | 36.7 | 0.1 | 6.3 |
| Subject Similarity | 0.6 | 61.3 | 0.8 | 38.8 | 0.9 | 4.7 |

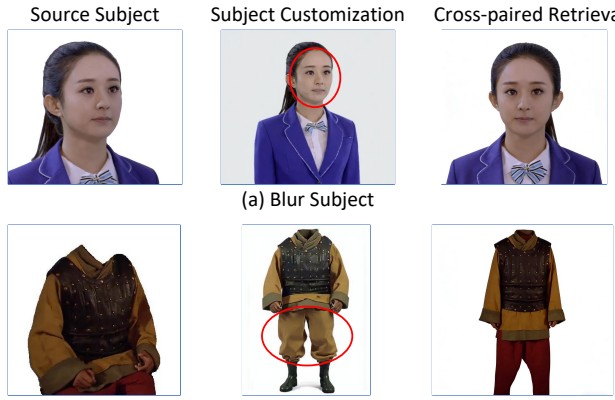

Source Subject  Subject Customization  Cross-paired Retrieval

(a) Blur Subject

(b) Inconsistent Identity

*Figure 11.* **Failure case of subject customization.** When applying image customization methods on a large scale dataset, the generated subject images sometimes tend to be blurry or have inconsistent identities.

**Cross-Pair Subject Retrieval.** Directly using subjects segmented from videos as reference images can cause unnatural copy-paste artifacts between subjects in generated videos. To eliminate this issue, we attempted to use personalized image generation methods (e.g., Dream-O (Mou et al., 2025) and Flux-Kontext (Labs et al., 2025)) to generate identity-consistent variations of these subjects. However, these generated results often exhibit unnatural artifacts or identity corruption and make large-scale application impractical, as shown in Figure 11. Thus, we adopted cross-pair similarity matching. Specifically, we aggregate video clips from the same long video and perform subject CLIP (Radford et al., 2021) feature similarity retrieval between them. If similar subjects are found, the reference image of the original subject is replaced. The similarity score threshold for Cross-Pair Subject Retrieval was set to 0.8. This method ensures the same subject appears in different poses and contextual backgrounds while guaranteeing all reference images are derived from real footage, thereby enhancing dataset diversity and reducing annotation bias.

Since not all videos have cross-paired data, we combined videos with segmented reference images and those with cross-paired reference images to create the final MuSA-2M dataset. This mixed data helps the model better learn identity consistency, assists in modeling high-level identity semantics, and avoids simple copy-paste effects.

## D. Additional results

This section presents additional experimental results to further verify the effectiveness and superiority of our model, which serves as a supplement to the main experimental results. The following subsections include more comparative results with existing methods, additional visualization results of our model.

### D.1. More comparison results

To fully demonstrate the superiority of our model over existing methods, we conduct additional quality comparison experiments on the OpenS2V-Eval dataset and our test data in Section B.4. As illustrated in Figure 12 and Figure 13, in multi-subject scenarios, VACE struggles to generate all subjects with consistent identity; Phantom often neglects the identity information of partial subjects; Humo is prone to errors in subject quantity processing, generating duplicate subjects or those inconsistent with the prompt; Kaleido lacks the capability of facial perception and frequently missing the identity of human face. Furthermore, these methods

have all undergone bias correction on large-scale real-scene datasets, making them less effective at handling reference subjects with special styles such as oil painting texture. In contrast, the method proposed in this paper can stably generate high-quality personalized videos with consistent identity features in both single-subject and multi-subject scenarios.

### D.2. More visualization results

To intuitively demonstrate the performance of our model, we provide additional visualization results on the OpenS2V-Eval benchmark and our test set. These results span a diverse range of subject customization scenarios, including multi-human, multi-object, and human-object hybrid settings. The visualizations shown in Figure 14 and Figure 15 confirm that our model stably generates high-quality videos across all these multi-subject scenarios, highlighting its strong potential for practical downstream applications such as film production, virtual content creation and personalized animation design.

## E. Limitations

While our method has achieved notable progress in efficient multi-subject personalized video generation, it still has certain limitations. First, the MuSA-2M is constructed from open-source datasets and in-house e-commerce data, with a shortage of high-quality out-of-domain data, which impairs the model's ability to generate arbitrary high-quality multi-agent interactive videos. Second, although our sparse routing strategy can reduce computational resource consumption during inference and unlock the potential for large-scale (4 or more) multi-subject personalized generation, the absence of reference samples for large-scale multi-subject scenarios in the training data renders the model unable to accurately model thematic consistency and interaction rationality in such scenarios. We plan to collect a large-scale dataset of videos containing 4+ salient subjects to enable accurate modeling of complex group interactions in the future work.

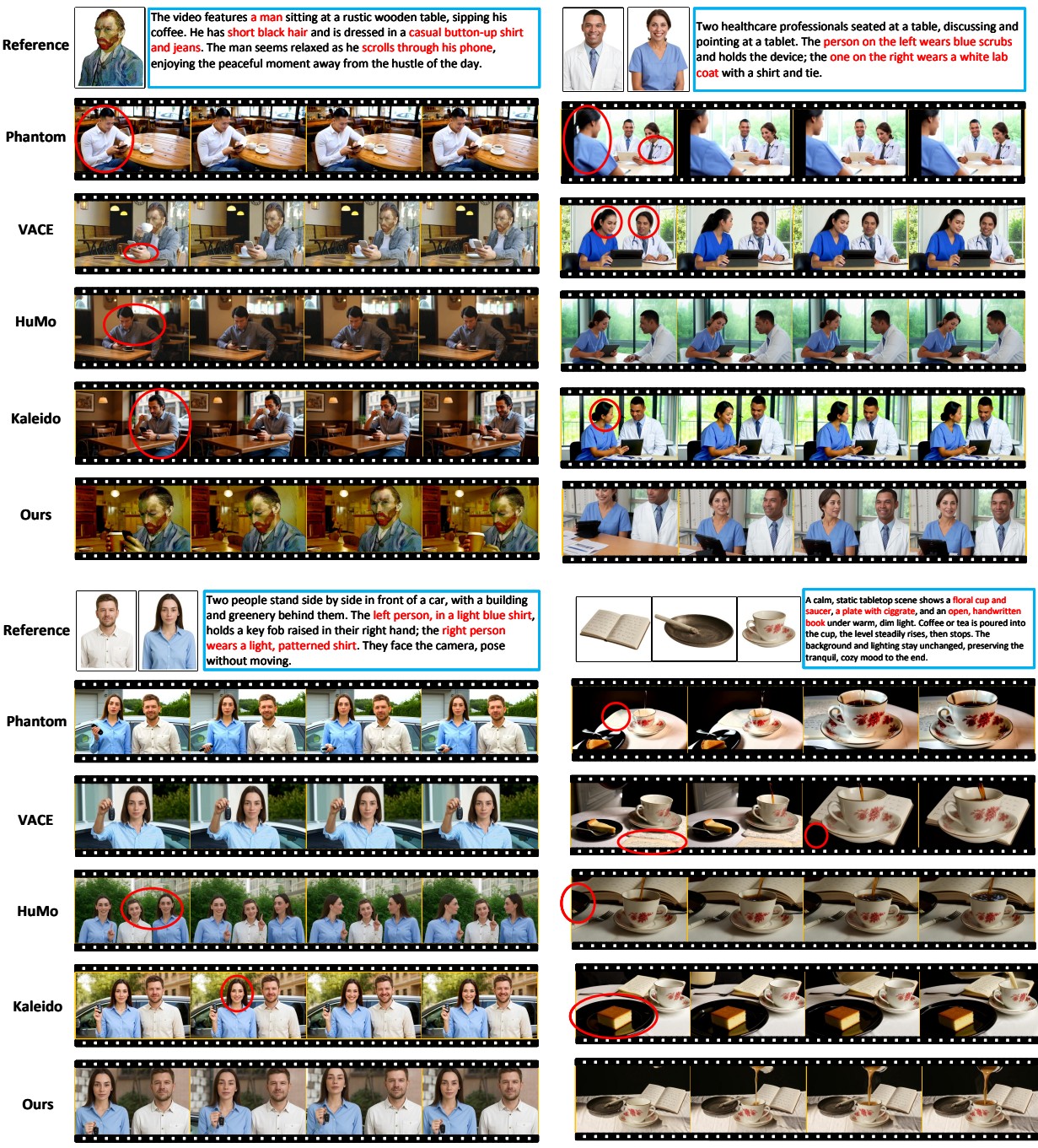

*Figure 12.* **More comparison on OpenS2V-Eval.**

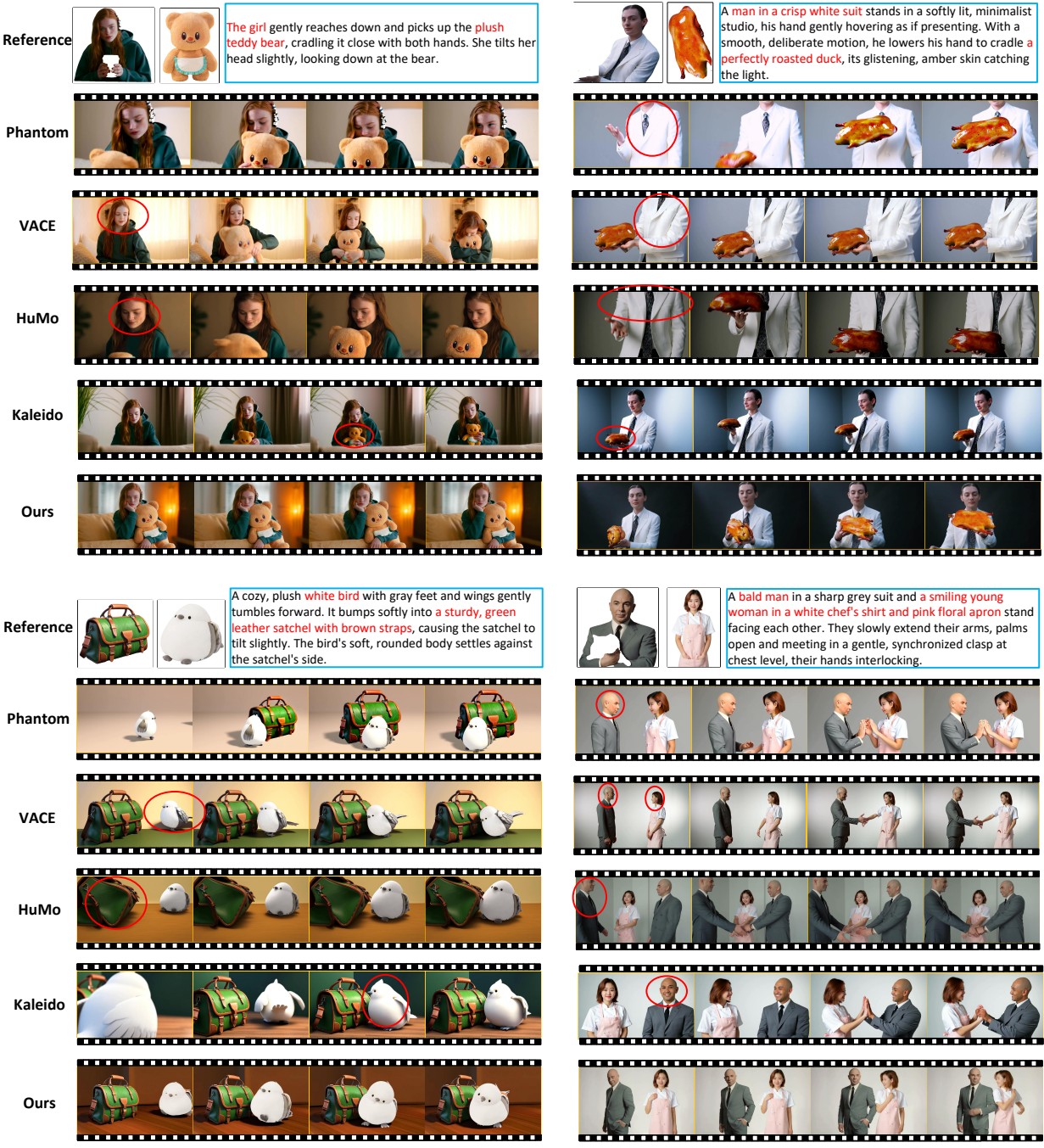

*Figure 13.* **More comparison on Our Test Set.**

A blonde woman in a black swimsuit stands at the water's edge.

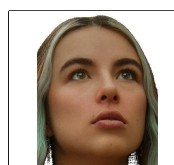 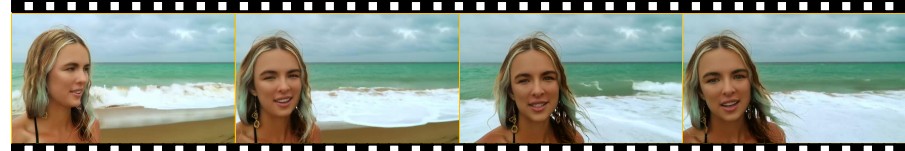

A man playing with his dog on the beach.

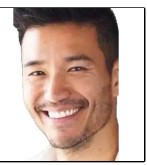 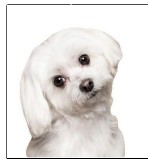 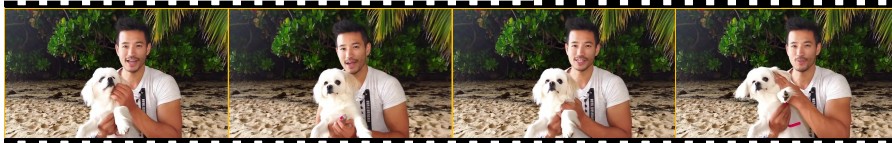

A man in a black t-shirt talk with a man in a light blue button-up shirt seated at a table.

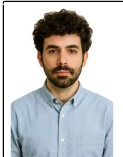 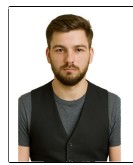 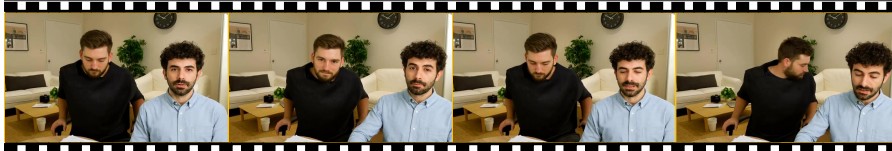

A woman in a white tank top and patterned shorts talking with a man in a plaid shirt.

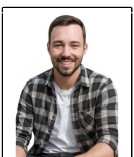 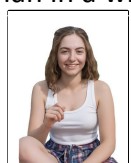 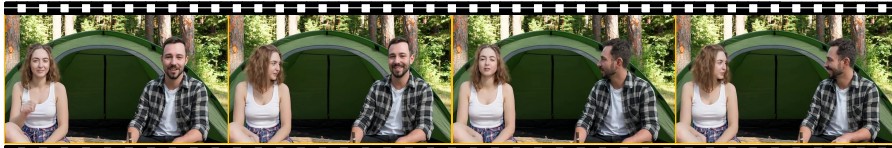

A black-and-white cat occasionally glancing around. A gray cat on the right watches it.

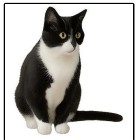 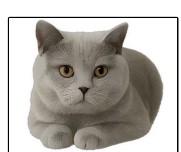 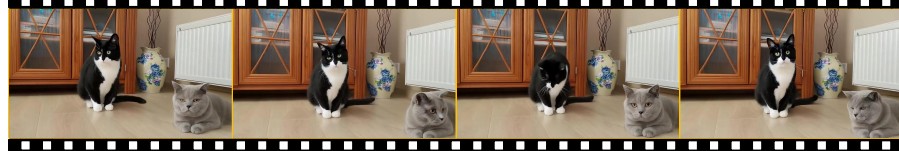

A group of women gather for a celebration. A woman in red holds a gift box.

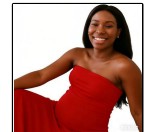 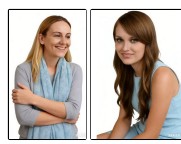 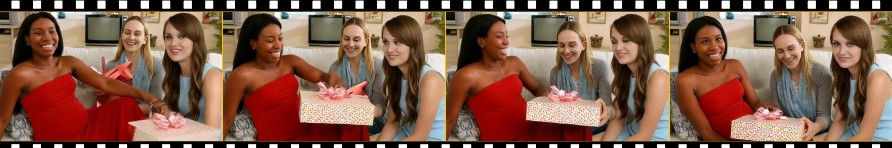

*Figure 14.* **More results on OpenS2V-Eval.**

A young woman in a grey sweater lifts a blue can of Snow Mountain beer.

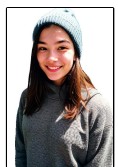 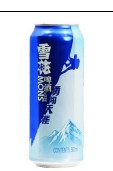 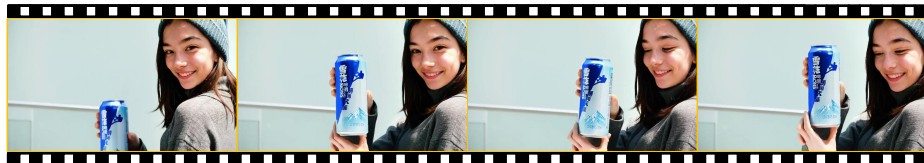

In a dimly lit bar, a cowboy was talking to a koala, while the koala was looking around.

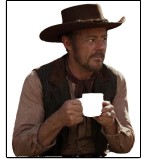 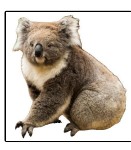 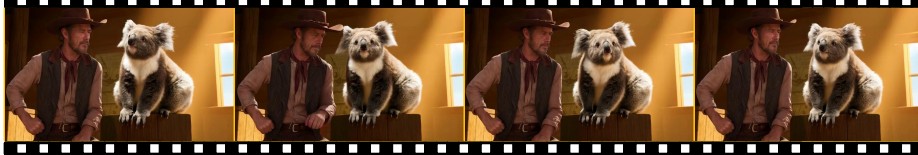

A man in the jacket and a woman in grey blazer raise their hands and touch togather.

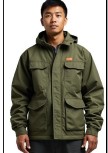 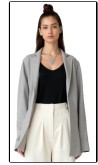 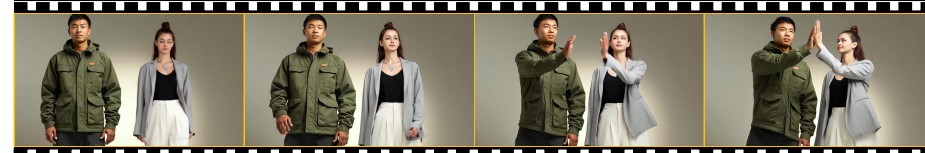

A fluffy Ragdoll cat gently paws at a mango, causing the fruit to tilt and roll slightly.

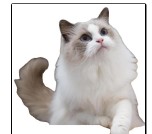 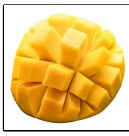 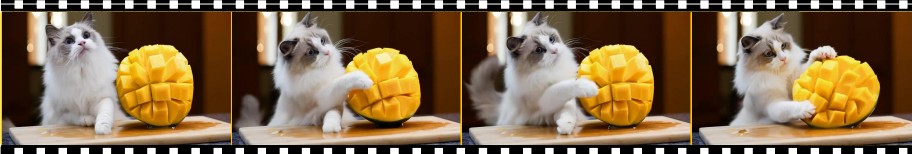

A Pikachu in a colorful pineapple-patterned shirt hold a green bottle of Snow beer.

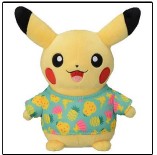 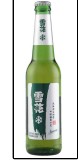 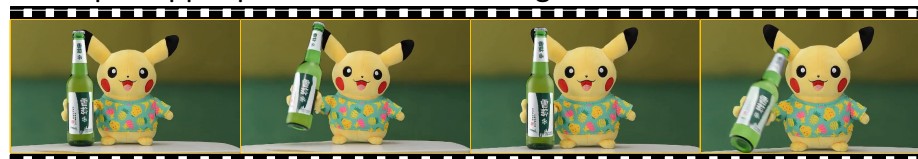

A coyote strides forward, turning its head to watch the koala perched on a low branch.

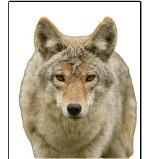 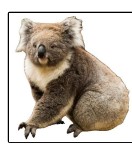 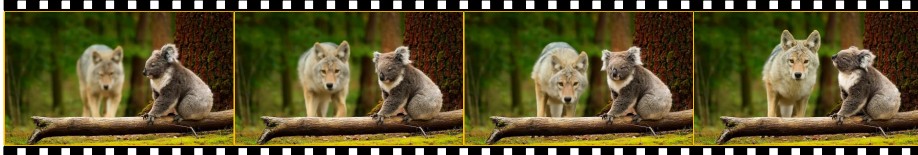

*Figure 15.* **More results on Test Set.**

