# OpenReview forum: "DiasR: Dual-Modal Identity-Anchored Sparse Routing for Efficient Multi-Subject Video Generation"
_ICML.cc/2026/Conference — ICML 2026 regular_

### Official Review · Reviewer_3t6G · 2026-03-04

**Soundness:** 3
**Presentation:** 3
**Significance:** 3
**Originality:** 3
**Overall Recommendation:** 4
**Confidence:** 3

**Summary:**

This paper addresses two major issues in multi-subject video generation: **identity drift/entanglement** and the **explosion of model computational overhead with the number of reference subjects**, while filling the gap of missing annotated datasets in the field. It proposes the DiasR framework, which integrates dual-modal identity anchoring and alignment with a sparse routing strategy, and constructs a large-scale annotated dataset to fill the data vacancy. Multi-dimensional experiments verify the advantages of the model in generation quality and computational efficiency. The paper also objectively analyzes the research limitations and potential social impacts, along with corresponding mitigation measures.

**Compliance With Llm Reviewing Policy:**

Affirmed.

**Key Questions For Authors:**

1. What is the empirical basis for presetting 10 learnable identity queries? Is there a quantitative ablation experiment on the impact of different query numbers on model metrics and computational efficiency?2. The choice of applying text-modal perceptual supervision to DiT layers 5–25 is based only on visual analysis. Is there a quantitative performance comparison for different layer ranges?3. Compared with other baseline methods, the characters and environments generated by this method seem to be disentangled—characters tend to be in the foreground with unnatural interaction with the environment. Is this a defect of the method caused by masking?4. The paper states that the method is unsuitable for scenarios with 4+ reference subjects, and all subjects shown in the paper and supplementary materials are no more than two. Are there also problems with 3–4 subjects (only quantitative results are shown in the paper)? Are there preliminary improvement strategies and corresponding experimental results for 4+ reference-subject scenarios?5. Videos are uniformly sampled to 81 frames. What is the reason for this choice? A 3-second video of 81 frames hardly supports large motions. What problems will directly applying this method cause for long videos with large motions (human/camera)?

**Limitations:**

Yes

**Strengths And Weaknesses:**

#### Strengths:The research is technically rigorous. The core modules are designed to target task-specific pain points with tight logical connections, and all core conclusions are supported by sufficient quantitative and qualitative experiments. The experimental design is comprehensive, covering baseline comparison, component ablation, efficiency analysis, and subjective evaluation, with metrics tailored to the task characteristics. The constructed dataset fills a key gap in the field. The proposed framework has both academic innovation and practical application value, and its core modules are originally designed for the task, providing new insights for efficiency optimization of related models. The paper is well-structured, with intuitive figures and tables, detailed supplementary materials, accurate literature positioning, and good reproducibility.#### Weaknesses:Some technical details and hyperparameter selections lack quantitative ablation experiments and are only determined based on empirical settings or visual analysis. The filtering thresholds and related modules in dataset construction lack detailed parameters and quantitative effect analysis. In addition, insufficient out-of-domain data limits the model’s generalization ability. There is no specific quantitative metric for performance degradation in multi-reference-subject scenarios, and the analysis of adaptability to special-style reference subjects is superficial. The selection of some loss functions lacks comparative validation. Furthermore, the description of a small number of core computing processes is relatively brief, and the impact of dataset annotation quality on model performance lacks quantitative verification.

---

> ### Author Rebuttal · Authors · 2026-03-30
>
> We highly appreciate the reviewer for positive recognition of our work, including its logically consistent module design, academic innovation, and the significance of the proposed dataset. We address each concern in detail below.
>
> **1. Number of Identity Queries**
>
> Identity queries are specifically designed to anchor distinct reference subjects. Crucially, these identity queries are not permanently bound to any specific entity. Instead, they only provide discrete identity anchors within each group of reference subjects, and can be abstracted as a form of subject positional encoding with an upper quantity limit. Accordingly, it suffices that the number of identity queries is no less than the number of subjects, and is irrelevant to model performance. In our experiments, we set this number to 10 as the upper limit of subjects, which is sufficient to cover the vast majority of practical application scenarios.
>
> **2. Quantitative Comparison for Different Blocks**
>
> Our base model comprises 40 Transformer blocks. Empirically, the middle blocks have stronger responsiveness to textual inputs. To further quantitatively evaluate the perceptual capability of different blocks toward the text modality, we employ the original Wan2.1-I2V-14B-480p model to generate videos from 100 reference images with one subject. The prompt was uniformly set to: a <subject> in <somewhere>. For each generated video, we extract the attention map of each DiT block and average them across 50 timesteps. We then compute the mean squared error between the normalized attention map and the ground-truth mask of the corresponding subject. The results demonstrate that blocks 6 to 25 achieve the optimal textual perception capability. While this range does not fully align with our pre-selected layers 5 to 25, it fundamentally corroborates our core intuition of selecting the middle blocks with the strongest representational capacity.
>
> |Blocks|1~5|6~10|11~15|16~20|21~25|26~30|31~35|36~40|
> |:-:|:-:|:-:|:-:|:-:|:-:|:-:|:-:|:-:|
> |**MSE**|0.216|0.084|0.112|0.093|0.098|0.121|0.146|0.175|
>
> **3. Unnatural Interaction with the Environment**
>
> This issue appears to occur only in certain scenarios, arising from the spatial constraints imposed by the subject mask. Such constraints cause the model to focus more on the subject than the background and tend to prioritize placing the subject in the foreground. This also leads to improved multi-subject consistency. However, as illustrated in the last row of Fig.15 (Supplemental PPT, Page 14), the model still successfully generates the effect of a subject gradually approaching from a distant environment. In future work, we will explore trade-off strategies between the mask constraints and natural environmental interaction.
>
> **4. Performance on 4 more subjects**
>
> To visually demonstrate subject consistency, most of our cases involve two distinct subjects. We present results with three subjects in the bottom-right panel of Fig.12 and the last row of Fig.14. In our experiments, both subject consistency and text alignment decrease when the number of subjects exceeds four. On one hand, our training data rarely contains more than four prominent subjects (e.g., four salient humans), causing the model to struggle with confusion when handling dense multi-subject scenarios. On the other hand, a larger number of subjects further increases the difficulty of sparse routing. We plan to collect a large-scale dataset of videos containing four or more salient subjects in future work.
>
> **5. Reason for 81 Frames**
>
> Our model is initialized from the Wan2.1-I2V-14B-480p foundation model, for which the recommended optimal frame count is 81. Constrained by the base model, we also adopt this setting. All baselines are similarly restricted to within 100 frames per single generation. Generating extremely long videos with far more than 81 frames in one pass may result in catastrophic temporal consistency collapse.
>
> **6. Detailed Parameters of Dataset**
>
> The detailed filtering thresholds are provided in Section C. The similarity score threshold for Cross-Pair Subject Retrieval was set to 0.8.
>
> These thresholds are solely for data cleaning and are not learnable hyperparameters for the model. Theoretically, higher filtering thresholds result in better data quality. Data quality and quantity exhibit a unimodal trend. Therefore, a reasonable trade-off between data quality and quantity is sufficient. The table below shows the percentage of clean data at some thresholds that are not mandatory. We conducted manual sampling after cleaning to ensure dataset reliability.
>
> ||Threshold (>)|Ratio (%)|Threshold (Ours) (>) |Ratio (%)|Threshold (>)|Ratio (%) |
> |-|:-:|:-:|:-:|:-:|:-:|:-:|
> |Aesthetic  Score| 5.0| 79.1|**5.8**|**32.3**|6.0|9.4 |
> |Motion Score| 0.005| 81.2|**0.01**|**61.2**|0.05|9.3|
> |Huamn Occupancy|0.04| 71.5|**0.1**|**52.3**|0.4|13.2|
> |Object Occupancy|0.01|63.2| **0.04**|**36.7**|0.1|6.3|
> |Subject Similarity|0.6|61.3|**0.8**|**38.8**|0.9| 4.7|

---

> > ### Author Rebuttal · Reviewer_3t6G · 2026-04-03
> >
> > The author's response aligns with my expectations, and the contribution of the paper is well-defined. However, the final outcome exhibits unnaturalness and imposes restrictions on the number of subjects. For this reason, I have decided to retain my original score.

---

> > > ### Author Response · Authors · 2026-04-07
> > >
> > > Thank you for acknowledging that the concerns are fully resolved and the contribution of our paper is well-defined.
> > >
> > > We note your remaining concerns regarding output unnaturalness and subject number constraints. These limitations mainly stem from the insufficient coverage of samples with four or more subjects in our training dataset. In the revised paper, we will clearly discuss this data limitation and plan to improve the model's capability for more subjects by expanding the multi-subject annotated dataset in future work.
> > >
> > > We will carefully address these points in the final version and greatly appreciate your valuable feedback.

---

### Official Review · Reviewer_5gxs · 2026-03-10

**Soundness:** 3
**Presentation:** 2
**Significance:** 2
**Originality:** 3
**Overall Recommendation:** 2
**Confidence:** 4

**Summary:**

This paper proposes DiasR, a framework for personalized multi-subject video generation that targets two key pain points: identity drift/entanglement across frames and quadratic inference cost as the number of reference subjects grows. The method introduces Dual-Modal Identity-Anchored Alignment, which uses learnable per-subject identity queries and ground-truth mask supervision to align visual self-attention and textual cross-attention with subject locations, and a Sparse Routing Strategy that dynamically assigns each video token to its most relevant subject and performs bucketed attention for efficiency. The authors also curate MuSA-2M, a large-scale multi-subject video dataset with subject-level masks. On OpenS2V-Eval, DiasR attains top identity-consistency metrics and exhibits nearly constant inference time with increasing numbers of subjects.

**Compliance With Llm Reviewing Policy:**

Affirmed.

**Key Questions For Authors:**

- Ablation Protocol Validity: Please explicitly clarify your ablation protocol. Are "w/o ID Query," "w/o L_mask," and "w/o Sparse" strictly internal ablations trained and evaluated under 100% identical settings as your main model, or are they borrowing numbers/architectures from Liu et al. (2025)? If the latter, you must provide true internal ablations.

- Training Setup Discrepancy: Reconcile the conflicting training details between the main text (10k steps / 64 A100s) and the supplement (12k steps / 64 A800s). What were the actual hardware, steps, batch sizes, and effective training compute used for the final model?

- Given the notable drop in Motion Amplitude with sparse routing, how do you justify the claim of high-quality video generation? Can you quantify the impact on perceived realism (e.g., FVD, user study) and discuss any architectural mitigations (like occasional full-sequence layers)?

- How exactly are “subject-related text tokens” identified and linked to identity queries? Please explain the process for complex prompts involving coreference or multiple mentions per subject. Is there a dependency on an automated text parsing step?

- Provide a rigorous formal definition for the visual similarity variable (S_i) in L_mask. How is it computed per block, normalized, and aggregated across time and space? Do gradients flow through the routing operation during training?

- Why was the system restricted to top-1 routing? Did you evaluate top-k or soft routing variants? Please provide data on how identity metrics and runtime trade-off as k increases.

- How does DiasR perform when the number of subjects scales beyond 6–7, particularly under heavy occlusion or physical interactions? Please provide failure case analyses.

**Limitations:**

The authors must explicitly acknowledge the negative impact of their sparse routing mechanism on motion dynamics (as evidenced by the drop in Motion Amplitude). The current manuscript treats this as a minor side effect, but suppressing motion to preserve identity is a significant limitation for a video generation model that needs to be critically discussed. Furthermore, the brittleness of the mean-pooled K_I descriptors in handling occlusions or identical twins/similar subjects should be documented as a limitation.

**Strengths And Weaknesses:**

Strengths:
- Logical Architecture Design: The bucketed attention mechanism is practically minded. Using a Gumbel-Softmax routing over pooled subject keys, combined with a null bucket to handle background tokens, is a sensible way to preserve cross-subject interactions while maintaining computational efficiency.

- Strong Evaluation Benchmark: The evaluation on OpenS2V-Eval is comprehensive, covering multi-human, human-object, and face scenarios. The runtime and FLOPs analysis effectively supports the authors' claims regarding the efficiency of the sparse routing mechanism.

- Valuable Dataset Effort: If the MuSA-2M dataset is released, it could serve as a solid resource for the community working on multi-subject video generation.

Weaknesses:

- Incremental Novelty and Missing Baselines: The proposed dual-modal supervision is essentially a straightforward extension of established attention-masking techniques already prevalent in image/video customization (e.g., CustomVideo, ContextGen). The paper fails to adequately compare against these closely related identity-alignment and attention-masking methods. Furthermore, the routing mechanism needs to be contextualized against other sparse token selection methods in ICC video DiTs (e.g., FullDiT2). Without these head-to-head comparisons, the technical novelty appears marginal.

- Flawed and Confusing Ablation Studies: This is a major concern. The ablation labeling (e.g., “w/o ID Query (Liu et al., 2025)” and “w/o Sparse (Liu et al., 2025)”) strongly implies that the authors are comparing their full model against external works rather than conducting clean, controlled internal ablations within the DiasR framework. It is currently impossible to verify if these ablations isolate the targeted components while keeping all other variables identical.

- Performance Degradation in Video Dynamics: The notable drop in Motion Amplitude under sparse routing is a non-trivial trade-off that is brushed over. A video generation model that suppresses motion dynamics to maintain identity consistency is heavily biased towards generating static or low-motion sequences, which fundamentally harms the perceived realism and utility of the model.

- Brittle Routing Mechanism: Relying on mean-pooled K_I descriptors for subject routing is an oversimplification. This coarse descriptor is highly likely to fail or cause misrouting in scenarios with high pose/appearance variability, shared fine-grained attributes among subjects, or heavy occlusions.

- Unfair Baseline Comparisons: The reported OpenS2V-Eval scores for baselines appear to rely partly on third-party evaluations or official protocols. It is unclear if these perfectly match the authors’ generation settings (e.g., sampling steps, guidance scale, sampler type), raising serious fairness concerns.

- Inconsistencies and Presentation Issues: The manuscript contains sloppy errors. The training setup is glaringly inconsistent: the main text claims 10,000 steps on 64 A100s, while the supplement describes a two-stage training totaling 12,000 steps on 64 A800s. Additionally, equation references are mismatched (e.g., the definition of visual similarity S_i vs. Eq. 4), and the mechanism for linking text tokens to identity queries is entirely underspecified.

---

> ### Author Rebuttal · Authors · 2026-03-30
>
> **1. Incremental Novelty and Missing Baselines**
>
> - #### **Attention Masking Techniques**
>
> DisaR is designed for multi-subject identity consistency, and vanilla attention masking alone cannot realize precise alignment between subjects, prompts, and videos, thus requiring dual-modal identity anchoring.
>
> CustomVideo imposes spatial constraints on textual attention maps and binds identity to textual tokens without visual constraints, leading to poor identity consistency; it works for animals and static objects but fails at precise human identity generation.
>
> ContextGen focuses on layout control for image generation by delimiting subject attention regions through predefined rules. However, it is hard to get precise temporal layouts that match prompts. Moreover, they do not establish a cross-modal alignment, which weakens the controllable capability of text over multiple subjects.
>
> DisaR addresses identity drift and entanglement in multi-subject video generation. We design learnable identity queries projected to dual-modal subject-specific anchors. Mask serves only as a supervisory signal, with core innovation in precise dual-modal subject representation anchoring. Additionally, visual masks are derived from subject similarity scores of video tokens via sparse routing, resolving identity entanglement.
>
> - #### **Routing Mechanism**
>
> FullDiT2 improves efficiency by compressing token-level redundancy in the video context.It uses an MLP to evaluate token importance via their Value, ignoring semantic associations between video tokens and specific subjects. This may lead to multi-subject entanglement.
>
> In contrast, DisaR proposes a dynamic subject-level sparse routing mechanism for binding video tokens to subjects. By computing the similarity between video tokens and each subject, it achieves one-to-one subject-level routing, which supports subsequent dual-modal identity alignment, ensuring deep identity perception across modalities. This mechanism reduces computational redundancy and mitigates identity entanglement.
>
> **2. Confusing Ablation Studies**
>
> Sorry for inserting the irrelevant references (to be removed in revision). All ablation studies were conducted in our framework by only removing the target component.
>
> **3. Video Dynamics**
>
> In Tab.2, sparse routing reduces MotionAmplitude but slightly increases NaturalScore, proving it does not compromise video realism. The top-1 routing eliminates chaotic motion from irrelevant subjects while preserving temporal consistency. DiasR has comparable MotionAmplitude to baselines but better FaceSim and NexusScore. User Study (Fig.11) confirms our superiority in video quality and naturalness.
>
> **4. Routing Mechanism**
>
> This is an effective strategy, while direct similarity computation is costly. Although mean pooling discards some details, it is sufficient for similarity matching. The aforementioned cases do not affect our representational capacity or lead to routing errors. Tab.1 shows our high FaceSim and NexusScore, verifying that mean pooling does not harm subject consistency.
>
> **5. Unfair Baseline Comparisons**
>
> The metrics of all baselines strictly follow the official standard settings of OpenS2V-Eval. We fully adopt the recommended settings of all baselines. OpenS2V-Eval only provides reference subjects, prompts, and metric computation.
>
> **6. Inconsistencies and Presentation Issues**
>
> Sorry for these writing errors. We will correct them. The total training step is 12,000. $S_i$ is the visual similarity score of the full video sequence, Eq. (4) is computed at a single index $ind$ in this sequence, with all indices concatenated in practice. For Gumbel-Softmax, we adopt the soft mode during training to preserve gradient flow, and the hard mode during inference. As stated in Line 199, the projected identity query vectors are added element-wise to the corresponding text token vectors, which are labeled by LLM.
>
> **7. Restricted to top-1 routing**
>
> Our top‑1 routing works at the subject-level rather than the token-level. In DiasR, each token in the video sequence interacts only with its most relevant subject, ensuring a one‑to‑one token-subject correspondence and avoiding identity entanglement. Extending to top‑k may introduce the risk of identity entanglement. Therefore, top‑1 routing is necessary.
>
> **8. Performance on 6-7 subjects**
>
> Over 6 subjects is an extreme case, which may cause subjects to be missing or entangle. This is related to the distribution of the dataset: statistically, most videos have 2 or 3 salient subjects. Incorporating training data with 4 more subjects will help alleviate this problem. When the input image is occluded, the model can complete the missing regions (Fig.14, 2nd row). The input is a face, while the output is a full body. For physical interactions, please refer to the 1st, 4th, and 5th rows of Fig.15 (and page 14 in the supplementary PPT), including holding a jar, shaking a mango, and grasping a bottle.

---

> > ### Author Rebuttal · Reviewer_5gxs · 2026-04-03
> >
> > Thank the authors for their response. My concerns have been partially addressed. However, i still have concerns about the novelty and video dynamics, which is important for my evaluation. I encourage authors to further refine their core method. I will keep my score.

---

> > > ### Author Response · Authors · 2026-04-07
> > >
> > > Thank you for your careful feedback and acknowledgment. We fully agree that novelty and video dynamics are important to this work. To clarify briefly:
> > >
> > > Our novelty lies in the explicit anchoring of distinct subjects to independent identity queries for drastically enhancing multi-subject consistency, and the sparse routing mechanism that elevates computational efficiency while disentangling multi-subject entanglement.
> > >
> > > For video dynamics, DiasR eliminates chaotic motion in videos. Quantitative experiments and user study demonstrate that the decrease in video dynamics does not affect the naturalness of our results.
> > >
> > > We will strengthen the presentation of novelty and video dynamics in the final version. We truly appreciate your constructive comments.

---

### Official Review · Reviewer_z8Dd · 2026-03-12

**Soundness:** 3
**Presentation:** 3
**Significance:** 3
**Originality:** 2
**Overall Recommendation:** 4
**Confidence:** 3

**Summary:**

This paper proposes DiasR, a framework for personalized multi-subject video generation. The method introduces a Dual-Modal Identity-Anchored Alignment mechanism that associates learnable identity queries with visual and textual features to enforce subject-specific consistency. In addition, the paper proposes a Sparse Routing Strategy that dynamically routes video tokens to the most relevant subject to mitigate identity entanglement and reduce computational overhead. Experiments demonstrate improvements in identity consistency and inference efficiency compared with prior approaches.

**Compliance With Llm Reviewing Policy:**

Affirmed.

**Final Justification:**

My concerns have been addressed. I also read other reviews and I would keep my current rating.

**Key Questions For Authors:**

Please see the weakness part.

**Limitations:**

yes.

**Strengths And Weaknesses:**

Strengths:
- The proposed dual-modal identity-anchored alignment provides a reasonable strategy for modeling subject-specific representations by associating identity queries with both visual and textual features.

- The sparse routing strategy is well motivated and aims to reduce both computational overhead and identity entanglement by dynamically routing video tokens to the most relevant subject.

- The experimental evaluation includes both qualitative and quantitative comparisons, showing improvements in identity consistency and inference efficiency.


Weaknesses:
- Identity queries seem an important component for constructing both textual attention and visual similarity maps. However, the paper does not clearly provide analysis on how performance changes when varying the number of queries. An ablation study on this design would help better clarify its impact.

- The designs for improving subject consistency and sparse attention have been explored in prior work on controllable video generation. The following recent works should be discussed, and the paper should better clarify its novelty beyond prior methods:
   - HiStream: Efficient High-Resolution Video Generation via Redundancy-Eliminated Streaming (CVPR findings 2026)
   - OneStory: Coherent Multi-Shot Video Generation with Adaptive Memory (CVPR 2026)
   - Scaling Zero-Shot Reference-to-Video Generation (CVPR 2026)

---

> ### Author Rebuttal · Authors · 2026-03-30
>
> We sincerely appreciate the reviewer for the positive feedback and for recognizing the advantages of our work: the rationality of the dual-modal identity-anchored alignment, the effectiveness of the sparse routing strategy, and the comprehensive experimental validation. The following summarizes all your concerns in-depth:
>
> **1. Number of Identity Queries**
>
>  Identity queries are specifically designed to anchor distinct reference subjects. Crucially, these identity queries are not permanently bound to any specific entity. Instead, they only provide discrete identity anchors within each group of reference subjects, and can be abstracted as a form of subject positional encoding with an upper quantity limit. Accordingly, it suffices that the number of identity queries is no less than the number of subjects, and is irrelevant to model performance. In our experiments, we set this number to 10 as the upper limit of subjects, which is sufficient to cover the vast majority of practical application scenarios.
>
> **2. Discussion with Other Works**
>
> We sincerely thank the reviewer for this valuable comment, and we acknowledge that these three works are all insightful and make remarkable contributions to consistency and efficient video generation. We will add a comprehensive discussion of these three works in the final version of our paper.
>
> - #### **On Sparse Attention Mechanism**
>
> HiStream focuses on the efficiency of high-resolution video generation. It achieves **token-level sparsity** by compressing spatiotemporal redundancy based on static priors, and actively prunes weak attention correlations via a sliding window. Its core essence lies in eliminating low-contribution components in attention computation.
>
> OneStory targets narrative coherence in multi-shot video generation, adopting semantics-guided **frame-level sparsity**. Its frame selection module picks the top-K semantically relevant frames from historical shots, and an adaptive regulator compresses the context. The sparsity operation concentrates on reducing redundant frames, without distinguishing between multi-subject identities.
>
> In contrast, our DiasR simultaneously addresses multi-subject consistency and the extra computational overhead caused by multiple subjects. It realizes **subject-level sparsity** through dynamic routing: each video token is routed to its most relevant subject, and the sparse structure evolves with the subject distribution. Furthermore, we firstly integrate sparse attention with multi-subject identity alignment, the sparsity operation directly serves to mitigate subject entanglement and isolate irrelevant subjects, thereby achieving superior multi-subject consistency.
>
> - #### **On Subject Consistency**
>
> The core goal of Scaling Zero-shot (Saber) is to break the dependency of S2V on dedicated triplet data and enable zero-shot subject consistency. It introduces random masks as dynamic references to simulate S2V training, and is primarily designed for general S2V generation.
>
> The core objective of our DiasR is multi-subject consistency control in S2V. We propose the first dual-modality identity anchoring mechanism, which, combined with learnable identity queries, enables the model to accurately track each subject’s identity at the semantic level. Meanwhile, our sparse routing mechanism further suppresses interference from irrelevant subjects, resolving identity drift, subject entanglement, and high computational costs in multi-subject scenarios.
>
> As evidenced by consistency metrics, DiasR outperforms Saber on both FaceSim (60.66% vs. 49.89%) and NexusScore (47.40% vs. 47.22%). The substantial improvement in FaceSim further validates that our identity anchoring mechanism reliably preserves fine-grained multi-subject identity consistency, such as facial features.

---

> > ### Author Rebuttal · Reviewer_z8Dd · 2026-04-03
> >
> > Thanks for the response. My concerns have been addressed. I also read other reviews and I would keep my current rating.

---

### Decision · Program_Chairs · 2026-04-30

**Decision:**

Accept (regular)

**Comment:**

### Reasons to accept

The paper tackles a challenging and timely problem in generating identity-preserving multi-subject videos. The proposed solution is technically sound, with reviewers particularly praising the practical-minded sparse routing strategy and bucket aggregation for reducing the overhead of disentangling individual identities from videos. The reviewers also agree on the rigor and sufficiency of the experimental validation of the proposed approach.

### Reasons to reject

The generated videos may contain unnatural motions, which points to some limitations of the generative pipeline. The proposed approach may also fail to work reliably for more than 6 subjects in a video, which the authors note is an extreme case that demands even more specialized machinery.

### Overall recommendation

The paper merits acceptance for presenting a clever approach to tackling a challenging problem, providing sufficient experimental validation, and clearly identifying its benefits and limitations. However, the work is primarily a proof of concept for a specialized problem and may be of limited interest to the broader ICML audience.